# Molecular fingerprinting of biological nanoparticles with a label-free optofluidic platform

Alexia Stollmann[1], Jose Garcia-Guirado [1], Jae-Sang Hong[2], Pascal Rüedi [1], Hyungsoon Im [2,3], Hakho Lee [2,3], Jaime Ortega Arroyo [1] ✉ & Romain Quidant [1] ✉

Label-free detection of multiple analytes in a high-throughput fashion has been one of the long-sought goals in biosensing applications. Yet, for all-optical approaches, interfacing state-of-the-art label-free techniques with microfluidics tools that can process small volumes of sample with high throughput, and with surface chemistry that grants analyte specificity, poses a critical challenge to date. Here, we introduce an optofluidic platform that brings together state-of-the-art digital holography with PDMS microfluidics by using supported lipid bilayers as a surface chemistry building block to integrate both technologies. Specifically, this platform fingerprints heterogeneous biological nanoparticle populations via a multiplexed label-free immunoaffinity assay with single particle sensitivity. First, we characterise the robustness and performance of the platform, and then apply it to profile four distinct ovarian cell-derived extracellular vesicle populations over a panel of surface protein biomarkers, thus developing a unique biomarker fingerprint for each cell line. We foresee that our approach will find many applications where routine and multiplexed characterisation of biological nanoparticles are required.

Accurate reconstruction of heterogeneous biological nanoparticle populations demands methods that satisfy three key parameters: sensitivity, high-throughput, and molecular fingerprinting. Extracellular vesicles (EVs), membrane-bound particles secreted by cells of all kinds[1,2], are a prime example of nanoparticle systems that would greatly benefit from characterisation methods that simultaneously comply with these three requirements. This is because the smaller the size of a particle, the greater the demand on sensitivity, which usually is paid in the currency of throughput. Similarly, the greater the number of biomarkers to screen, the lower the throughput. Thus, the ideal approach would be one which can detect these biological nanoparticles at the single particle level regardless of size in aqueous environments, and sample a statistically significant number of

particles (>10,000 events) within a reasonable time, i.e. on the time-scale of minutes to an hour. Lastly, the approach should differentiate between subpopulations expressing relevant biomarkers and minimise the rate of false positive. To date, fluorescent-based single-particle assays are the most established and prevalent due to their intrinsic specificity, single-molecule sensitivity, and compatibility with microfluidics. Fluorescence-based molecular fingerprinting has so far been achieved through either sequential read-out of different fluorescent probes[3,4], spectral emission decoding[5], spatial patterning[6], or a combination thereof[7]. Despite widespread use, fluorescence-based detection has intrinsic limitations either in the form of labelling efficiency, fixed photon budget or labelling incompatibility[8]. As a result, there is a need for all-optical label-free alternatives compatible with

[1]Nanophotonic Systems Laboratory, Department of Mechanical and Process Engineering, ETH Zurich, 8092 Zurich, Switzerland. [2]Center for Systems Biology, Massachusetts General Hospital, Boston, MA 02114, USA. [3]Department of Radiology, Massachusetts General Hospital, Harvard Medical School, Boston, MA 02114, USA. ✉e-mail: jarroyo@ethz.ch; rquidant@ethz.ch

high throughput microfluidics that can deliver all the benefits of single-molecule fluorescence assays without the constraints associated with labelling.

From the available all-optical label-free methods, those based on elastic scattering, and particularly those belonging to the family of digital inline holography, have become one of the most promising, as they now routinely achieve detection sensitivities down to the single protein[9–11], nucleic acid[12], and micelle level[13] that rival single-molecule fluorescence. Similarly, these approaches are uniquely suited to in situ characterise different surface functionalisation strategies[14,15]. Yet their translation to routine particle characterisation faces challenges related to high throughput detection of multiple specific analytes. To understand this, it suffices to consider that the amount of light scattered from said biological nanoparticles pales in comparison to light scattered by the substrate roughness; thus, the extreme sensitivity of all these surface-based techniques hinges on an imaging modality whereby the static background from the observation area is constantly updated. Such imaging modality, termed in some cases as differential imaging, restricts the assay to single, relatively small fields-of view (FOV), which rarely exceed the scale of 100 s of $\mu m^2$. These restricted sensing areas aim to minimise deleterious effects from either parasitic background scattering from the imaging optics or unwanted interferences due to the coherent nature of the light source typically used. For systems that do not demand the highest sensitivity, i.e. in the absence of differential imaging, the scattering signal from the substrate roughness as well as unwanted interferences that may arise from multiple interfaces, a common scenario in microfluidic devices, set the lower limit of detection and should be minimised throughout any surface functionalisation step. Although the substrate roughness can be significantly reduced using atomically flat substrates like mica[16], this comes at the expense of losing target specificity. Conversely, functionalising the surface with capture probes, e.g., antibodies or aptamers, introduces specificity, but also increases the substrate roughness. Even in the few cases where multiplexed detection in a non-differential imaging mode has been achieved, samples are imaged in air rather than in their native aqueous environment to enhance the scattering contrast[17].

Microfluidic integration can address the throughput and multiplexing challenges; nevertheless, finding a functionalisation scheme that delivers target specificity and minimises non-specific binding without introducing additional unwanted scattering signals remains an important obstacle. Despite the availability of numerous strategies, the harsh conditions involved in some steps in the assembly of PDMS-based microfluidics, such as plasma treatment followed by baking at high temperatures, compromise the integrity of any functionalisation. A recent alternative has been demonstrated by using masks during the assembly process; however, the need for μm-level alignment between a chip and the protective element, in the case of complex chip designs, imposes a steep technological restriction[18]. As a result, these state-of-the-art functionalisation approaches are incompatible with complex PDMS microfluidics, and in situ/on-chip solutions should be sought. Nevertheless, existing in situ alternatives require either long incubation periods on the timescale of hours, as is the case of poly(ethylene) glycol (PEG)-based strategies; or compromise on the degree of passivation, for instance, bovine serum albumin[19].

Bringing together state-of-the-art, all-optical label-free approaches with microfluidics requires an integrated solution that addresses the limitations intrinsic to each tool. In this work, we present a label-free optofluidic platform that delivers high throughput molecular fingerprinting solution for characterising heterogeneous nanoparticle samples. Specifically, we first identified a surface functionalisation protocol, in the form of high-quality supported lipid bilayers (SLBs), that acts as a building block to integrate microfluidic technology with label-free detection with single-particle sensitivity. Using this building block, we implemented a label-free immunoaffinity pull-down assay

and, in situ, assessed the performance of each stage of the functionalisation by taking advantage of the highly sensitive and label-free detection scheme of the platform. Finally, we showcase all the features of the platform by profiling populations of EVs from four different ovarian cell lines with single EV sensitivity using a panel of surface biomarkers from which we generate characteristic fingerprints for each EV subpopulation.

## Results and discussion
### Concept and experimental workflow
In this work, we fulfilled the requirements for label-free molecular fingerprinting of heterogeneous nanoparticle suspensions by focussing our efforts around three main concepts: (i) large FOV imaging with single particle sensitivity, (ii) high throughput, small volume, and individually addressable microfluidic channels and (iii) an in-chip surface functionalisation protocol for pull-down immunoaffinity assays (Fig. 1).

For large FOV imaging, we used an inline holographic microscope in reflection geometry with an intrinsic requirement of a spatially incoherent light source as we are only interested in interferometric contributions between the surface and nanoparticles immobilised to it. Figure 1B schematically depicts the optical read-out strategy. As an imaging area, we targeted illumination FOVs on the order $100 \times 100\ \mu m^2$, which are rarely achieved with interferometric scattering (iSCAT) microscopy, a digital inline holography approach, with high numerical aperture (NA) objectives due to the presence of detrimental parasitic fringes that arise from the reflections from multiple closely spaced interfaces in microfluidic chips. In addition to reducing these parasitic interferences when imaging through microfluidic chips[20], the spatially incoherent illumination drastically reduces the influence of speckles. To do so, the output from a narrowband fibre-coupled light emitting diode (LED) was relay imaged onto the sample. Light scattered by the sample, as well as the weak reflection from the substrate interface, was collected by the high NA objective, and subsequently, their interference was imaged onto a camera. Such illumination scheme is not limited to LEDs as a similar performance was also obtained by reducing the spatial coherence of a diode laser with a combination of a rotating ground glass diffuser and a multimode fibre (Supplementary Fig. 1). To extend the FOV, we followed established computer vision routines to stitch a series of raster scanned images.

To satisfy the low volume reagent, multiplexing, and throughput requirement, we used PDMS microfluidic technology based on Quake microvalves[21] (Fig. 1C). These chips were composed of a control (orange) and flow layer (light blue) to independently address different sensing channels (black arrows), and finely control each step of the immunocapture assay without interference from the user. Here each channel represented a different experiment programmatically controlled via a computer interface, thereby opening the possibility for long-term automation. To maintain uniform flow conditions all channels were designed with the same microfluidic resistance by keeping the dimensions of each channel fixed. Uniform flow rates are critical to guaranteeing consistent advection-driven kinetic conditions and minimising mass-transport limited effects throughout all the assays within a chip. In terms of total volume, each sensing area corresponded to 10 nL (length, width, height: 3 mm, 0.3 mm, 0.01 mm), which upon including the inlet and outlet path lengths, increased to ~40 nL per channel. Added together, the whole microfluidic device operated with less than 0.5 μL of sample.

For surface chemistry and to bring both established microfluidics and imaging technologies under a common umbrella, we opted for SLBs (Fig. 1D) as the basis for the in-chip functionalisation protocol due to their biomimetic nature, ease of preparation, intrinsic anti-fouling properties, and on-chip compatibility[22–24]. The SLBs prepared by fusogenic-assisted vesicle fusion simultaneously acted as a passivating coating against non-specific binding, and as a building block for the

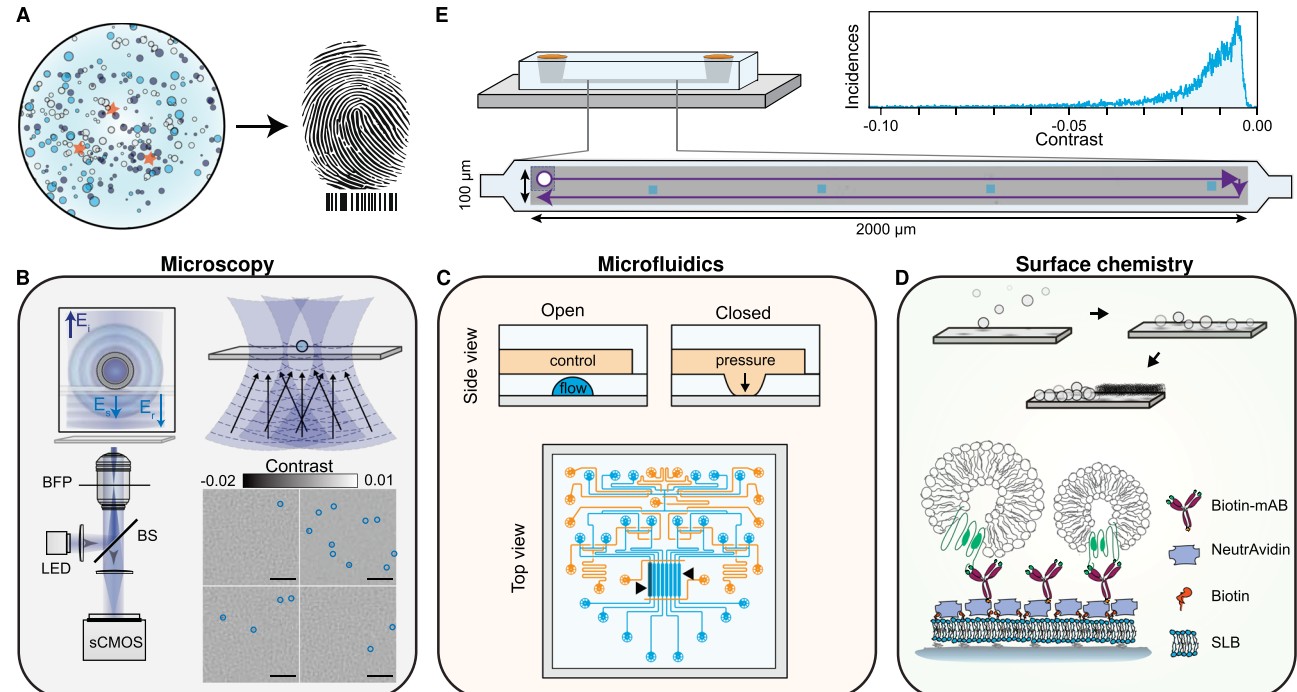

**Fig. 1 | Concept and workflow of the label-free optofluidic platform.**
**A** Conceptual illustration of the aim of the platform. The platform is based on three main toolboxes. **B** Microscopy toolbox: schematic of the optical system for large FOV imaging with single particle sensitivity based on spatially incoherent inline holography in a reflection geometry together with four representative zoomed-in images with diffraction-limited spots identified with blue circles. Inset: the working principle relies on detecting the interference between the weakly scattered light from the sample, $E_s$, and the reflection from the substrate/water interface, $E_r$. Scale bars: 5 μm. **C** Microfluidic toolbox: representative two-layer microfluidic chip design composed of a network of valves (orange) and flow channels (blue). The black arrows highlight the section of independently addressable channels used for sensing. **D** Surface chemistry toolbox: schematic representation of the in-chip functionalisation scheme based on SLB formation by liposome fusion, which acts as the building block for the immunoaffinity pull-down assays. **E** Workflow of the platform: representative experimental image scan of a sensing channel obtained by stitching multiple fields-of-view together with the resulting contrast distribution of all localised single particles. The scattering contrast signals are retrieved upon localising all the diffraction-limited spots above a signal-to-noise ratio (SNR) threshold, an example shown in (**B**).

immunocapture pull-down assay. Regarding the pull-down functionalisation scheme, NeutrAvidin molecules coupled biotinylated monoclonal antibodies to the SLBs doped with biotinylated lipids.

As a general workflow to either evaluate the performance at each stage of the functionalisation process or molecularly fingerprint EV populations, we raster scanned the sample along an area covering ~66% of the microfluidic sensing channel length (3 mm long) and stitched the acquired images together to generate image scans such as Fig. 1E. After flat-field correction, diffraction-limited spots (corresponding to biological nanoparticles, surface inhomogeneities or defects) were localised, their signal contrast obtained and subsequently plotted to determine their contrast distribution (Fig. 1E). These large image scans allowed us to build robust statistics, increase throughput and identify inhomogeneities in the surface functionalisation protocol. Furthermore, we opted for label-free imaging over other complementary approaches, such as fluorescence and AFM, because it provides an in situ quantitative characterisation of the very same sensor area at each stage of the immunoassay, which simply would not be measured via other approaches; thus, enabling us to account for slight differences in SLB quality between experiments.

### Supported lipid bilayers (SLBs) as the building block for immunoaffinity assays

The quality of the sensing substrate is of utmost importance for any label-free assays, as unwanted scattering from imperfections or defects will contribute to a false positive readout. This is particularly critical in the case of SLBs if one considers the potential overlap in sizes between EVs and any remaining unruptured liposomes from the SLB formation[25–27]. Even in the case of sensing based on differential

imaging[9–11,28], the presence of considerable scattering signals, such as large unruptured liposomes, impose tighter experimental constraints in the form of better sample stabilisation to compensate for the minute sample drifts that push the differential imaging approach away from the shot noise limited detection. To determine the most suitable lipid coating strategy, i.e. one that effectively reduces the likelihood of false positives during a sensing assay with high reproducibility, we screened different SLB preparation methods. As a metric, we aimed to minimise the number of scattering signals present in the formed bilayer.

For the SLB formation, we chose the fusogenic agent-assisted bilayer formation strategies, as they are the most promising in generating high-quality continuous lipid coatings with minimal defects irrespective of lipid composition and substrate properties[29,30]. We specifically used the α-helical (AH) peptide as the fusogenic agent since buffer washes can fully remove it from the formed bilayer, and, therefore, not influence further downstream steps[29]. The fusogenic activity of the AH peptide depends on the membrane curvature of the unruptured liposomes, and thereby their size; with smaller liposomes having higher curvature and peptide activity[31,32,33]. To determine the liposome size distribution that leads to the most reproducible and suitable bilayer for label-free sensing, we tested different preparations based on either extrusion or bath sonication. We monitored the formation of the SLBs with emphasis on three key stages: the bare substrate in the presence of buffer solution (PBS), the initial bilayer formed after liposome fusion (liposome), and the final bilayer after AH peptide incubation and subsequent buffer rinsing (peptide) (Fig. 2A). One of the key advantages of this functionalisation scheme, when combined with microfluidics, is the speed of preparation, which occurs on the timescale of minutes (Supplementary Movies 1, 2).

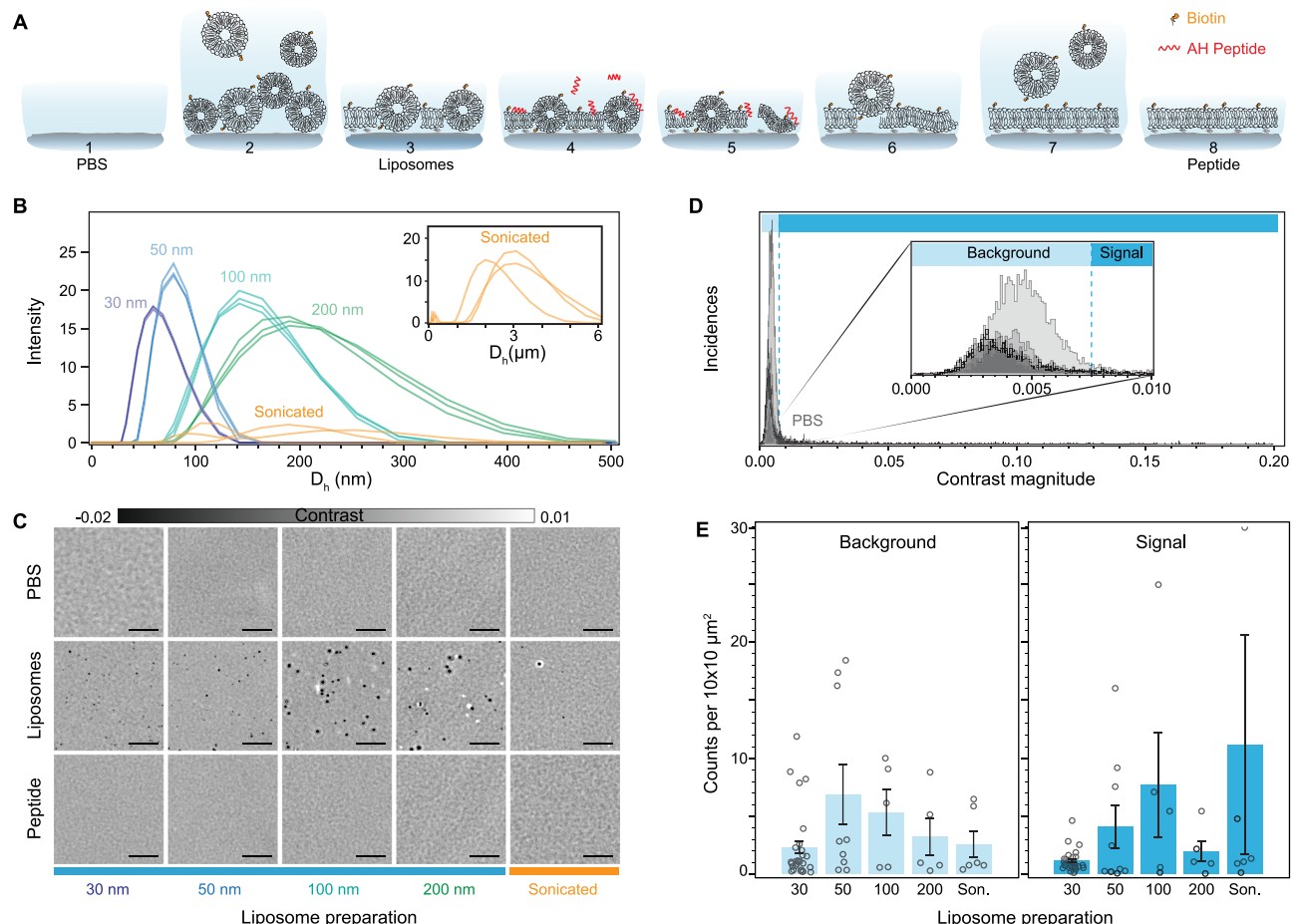

**Fig. 2 | Characterisation of the number of bilayer defects. A** Diagram showing the steps involved in the preparation of the supported lipid bilayer via fusogenic AH peptide interaction and osmotic stress. **B** Hydrodynamic size of the different liposome preparations as determined by dynamic light scattering. **C** Zoom-in of representative images for each preparation method at the different stages of the peptide-mediated supported lipid bilayer formation process. PBS: clean substrate exposed to only buffer solution; Liposomes: substrate after vesicle fusion and buffer rinsing; Peptide: supported lipid bilayer after peptide incubation and osmotic shock buffer rinsing. **D** Particle contrast histograms from all localisations found in substrates exposed only to PBS solution ($N = 6$). Each curve with the same transparency level and grey intensity corresponds to an approximate scanned area of 0.2 mm². The overlap between different curves is indicated by the different degrees of transparency. **E** Particle localisation density as a function of SLB preparation obtained from an approximate scanned area of 0.2 mm² which is categorised according to the contrast falling within background and signal regions, respectively. Data were expressed as mean ± SEM over independent over $N = (28,9,5,5,6)$ independent channel scan replicates corresponding to different functionalised surfaces. Scale bars: 5 μm.

For extrusion, the polycarbonate membrane pore size tuned the liposome size from 30 to 200 nm, whereas bath-sonication offered a minimal sample preparation at the expense of no control over the size distribution. Their respective size distributions determined by dynamic light scattering are shown in Fig. 2B, with liposomes prepared via extrusion displaying higher uniformity and reproducibility compared to bath-sonicated ones. Figure 2C shows representative zoom-in images, corresponding to an area of 20 × 20 μm² for each of the stages, to highlight the differences in SLB formation driven by substrate-vesicle interactions, as well as between peptide-induced bilayer repair. The first row, corresponding to the buffer-only step, provides an initial quality assessment of the cleaned glass substrate. At this stage of the process, we observed the presence of substrate roughness together with the inherent inhomogeneity of the substrate, either in the form of defects or contaminants. The second row shows representative examples of the formed SLB after vesicle fusion and buffer rinsing to remove excess liposomes. Here, the effect of substrate-liposome interaction is most noticeable in the number and signal contrast of the diffraction-limited spots. These diffraction-limited spots were assigned as either membrane defects in the form of unruptured liposomes, trapped liposomes, and inhomogeneities in the bilayer, or

defects already present in the bare substrate. Qualitatively, SLBs formed via liposome fusion alone favour larger liposome preparations (200 nm and sonicated), as they are more likely to rupture spontaneously compared to smaller ones[14]. The third and final row shows the bilayer after continuously flowing in the fusogenic peptide, followed by an osmotic shock upon buffer exchange. The osmotic shock is not due to the peptide, but rather by the differences in ionic strength between aqueous solutions prior to after incubation with the AH peptide. In all cases, the bilayers treated with the AH peptide significantly reduced the number of membrane defects compared to those formed by liposome fusion alone. To assess the quality of the final bilayer, we determined the number of defects before and after bilayer formation within each sensing channel, and reported them in the form of density, i.e., counts per area of 10×10 μm². We chose this area to allow meaningful comparison amongst most state-of-the-art label-free detection schemes, which have FOVs with dimensions ranging in the tens of microns[28,34,35]. To do so, we performed image scans covering an area of approximately 0.2 mm² over a minimum of three different substrates for each liposome preparation. Here we assign defects to any diffraction limited signal that is 4× the noise floor. This SNR cut-off was selected to minimise the occurrence of false positives

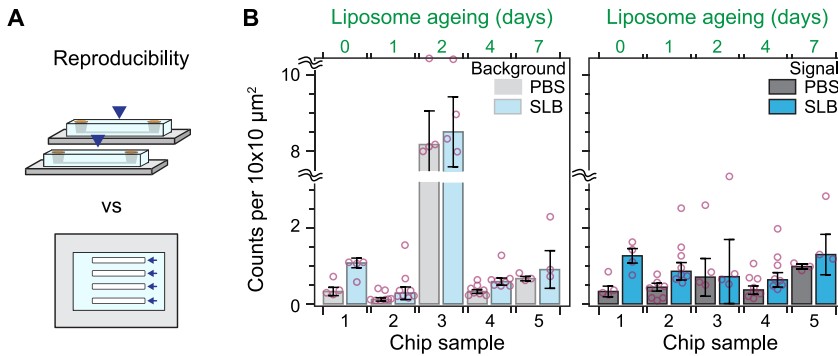

**Fig. 3 | Robustness of the supported lipid bilayer. A** Cartoon depicting the difference between chip-to-chip (inter-) and within-chip (intra-) variability. **B** Number of defects within the expected background and signal contrast regions for different chips, before and after SLB formation. The number of days after liposome preparation for each chip appears on top. Data were expressed as the median of scans of an approximate area of 0.2 mm² ± SEM over $N$ independent channels within each chip ($N = 4,8,4,8,3$).

attributed to noise fluctuations. As a first step, we determined the baseline contrast distribution of defects present in the bare substrate with buffer. Fig. 2D shows that most defects fall within a narrow contrast range, with a cut-off contrast value of $7.5 \times 10^{-3}$ as indicated by the dashed vertical line. We attributed these to surface roughness and minute substrate inhomogeneities. We classified this narrow contrast region as background, and the one above this threshold as the signal. This classification effectively minimises the influence of substrate roughness and small inhomogeneities in the defect density metric without the need for further image processing, i.e., background subtraction or differential imaging.

Across the different SLB preparations, we observed similar defect occurrences in the background region as well as the potential to form low-defect bilayers in the signal region (Fig. 2E). However, in the detection sensing window, liposomes prepared via extrusion with a 30 nm polycarbonate pore size formed the most reproducible SLBs, reflected by the smallest average defect density. Although the other approaches led to potentially high-quality lipid coatings, as shown in Fig. 2C, there was a high level of variability when assessed over larger observation areas, stressing once again the importance of large FOV imaging. We can rationalise these results by considering the interplay between two complementary size-dependent liposome rupture mechanisms: spontaneous rupture, with the rate of rupture increasing with size, and peptide-induced rupture, which due to the membrane curvature sensitivity of the AH peptide, preferentially ruptures liposomes with diameters below 125–150 nm. The size distribution from Fig. 2B confirms that the 30 nm liposome preparation has the smallest fraction of liposomes above the AH peptide size cut-off, and thereby does not depend on the spontaneous rupture mechanism. As a corollary, the probability of having a higher proportion of unruptured liposomes unaffected by the AH peptide is significantly higher for all other preparations and now becomes dependent on the vesicle-substrate interactions, thereby leading to a greater variance.

To put our results in the context of other works, our lowest defect density per 100 µm² is on the order 1.0 compared to the 0.05 counts previously reported[30]. The higher sensitivity and the label-free nature of our imaging platform can account for this discrepancy by considering analysing much larger areas makes it more statistically likely to find substrate defects and inhomogeneities.

**Microfluidic chip reproducibility and robustness**

Inter- and intra-chip reproducibility are critical for microfluidic-based assays. We assessed the reproducibility of the SLB formation process over multiple microfluidic chips (including different designs), and across different portions of the substrate. The microfluidic chips were designed to have between 4 to 8 independent flow channels, from which an imaging area equivalent to 0.2 mm² was scanned and the corresponding defect density determined. Figure 3 shows that across five independent chips measured over different days after liposome preparation, the defect density in the signal region is highly reproducible both within and across microfluidic chips. Moreover, the presence of high background signal levels from the bare glass substrate, e.g. chip 3, did not affect this high degree of reproducibility in the signal area. Also, despite the number of defects in the signal region largely correlated with the underlying quality of the bare substrate, the median remained near the value reported in Fig. 2E.

As the liposome sample aged, we observed a decrease in the spontaneous rupture frequency alongside an increase in unruptured liposomes prior to peptide treatment (Supplementary Fig. 2), in agreement with similar experimental work[36,37]. In both these prior works, the decrease in rupture frequency associated with liposome ageing was correlated with a decrease in the size of the liposome population. Besides a lower spontaneous rupture rate with decreasing liposome size, Cho et al. proposed a model based on time-dependent changes in the liposome structure to explain the higher amount of unruptured liposomes[36]. In this model, structural relaxation of the liposomes from ellipsoidal to spherical-shaped particles lowers the interaction strength between the liposome and the substrate, resulting in a higher energetic barrier for spontaneous rupture. Nonetheless, upon peptide treatment, there was no significant defect density dependence on liposome ageing (Fig. 3B, one-way Anova: $P = 0.534$). These results highlight the robustness and flexibility of the platform to reagent ageing. This feature allows the decoupling of the liposome preparation steps from the bilayer formation ones; a critical aspect when dealing with microfluidic devices.

**Compatibility with on-chip immunoaffinity capture strategies**

To access the standard immunocapture functionalisation scheme based on NeutrAvidin as a linker between biotinylated antibody and lipid, all bilayers were composed of POPC: biotin DOPE lipids in a 99:1 molar ratio. At this molar ratio, we expected an almost complete antibody coverage of the substrate, given an estimated surface density of 1.4 biotins per $10 \times 10$ nm² (2.3 pmol/cm²), slightly below the minimum doping to achieve a full monolayer of NeutrAvidin previously reported to occur at 2.8 biotins per $10 \times 10$ nm² (3.5% molar biotin, 8 pmol/cm²)[38]. We chose POPC as the main phospholipid component in our liposome preparation based on its favourable physical properties for SLB formation, namely: zwitterionic nature, low melting temperature, preference to form lamellar rather than hexagonal structures, and cylindrical shape with little to no curvature[39–41].

To evaluate whether progressive functionalisation steps, i.e. the addition of NeutrAvidin followed by biotinylated antibodies, impact

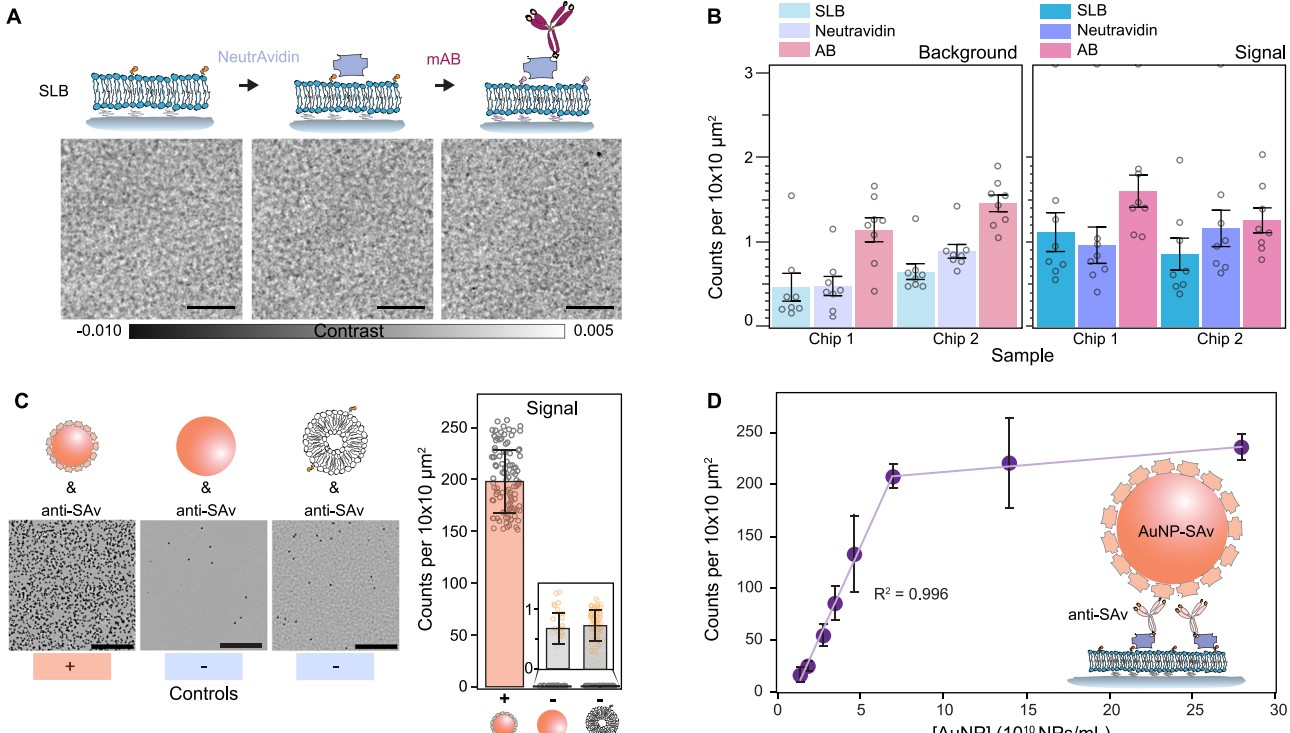

**Fig. 4 | On-chip immunoaffinity capture assay validation. A** Representative zoom-in images of the substrate after each functionalisation step: bilayer formation, NeutrAvidin incubation, and biotinylated antibody incubation. Scale bars: 5 µm. **B** The number of localisations after each functionalisation step per independent channel within a chip ($N = 8$), represented as mean ± SEM. **C** Validation of the immunoaffinity functionalisation using streptavidin-functionalised AuNPs (AuNP-SAv) as a positive control, and carboxylated AuNPs and biotinylated-liposomes as negative controls. The bar plot represents the median NP density recorded per single FOV over an area scan ± SD over $N$ independent FOVs ($N = 102$, 21, 102). Scale bars: 10 µm. **D** Dose response for different concentrations of streptavidin-functionalised 20 nm gold nanoparticles. Each data point corresponds to the mean of scans covering an area of 0.2 mm² ± SD over multiple different and independent channels within each chip ($N = 2$ chips).

the bilayer quality, we quantified the number of defects accumulated at each step (Fig. 4A). In the contrast region assigned to the background, we observed no major differences in the quality of the bilayer other than a small rise in the number of defects, which we associated to an increase in surface roughness caused by the respective randomly oriented protein coatings (Fig. 4B). The signal region exhibited a slight increase in localisations after antibody incubation, yet without statistical significance (one-way ANOVA: Chip 1, $P = 0.109$; Chip 2, $P = 0.303$), likely attributed to aggregates (Fig. 4B).

The principle behind the immunoaffinity pull-down assay was to target analytes located at the surface of a nanoparticle, irrespective of the nanoparticle internal composition and origin (i.e. biological or synthetic), and to immobilise the nanoparticle onto the substrate via antibody–antigen interactions. We therefore validated the immunoassay by first flowing a sample of streptavidin-labelled 20 nm gold nanoparticles (AuNPs-SAv) as positive control, and bare 40 nm gold particles and biotinylated-liposomes as negative controls against the antibody layer (Fig. 4C). The rationale for choosing AuNP-SAv as a positive control was that they represent a synthetically homogeneous nanoparticle population with well-defined physicochemical properties, which has been widely used to assess the performance of various interferometric-based label-free microscopes across different groups[16,42–44] thanks to their high SNR and low coefficient of variation size and therefore particle contrast. As expected, the AuNPs-SAv showed nearly 200-fold more binding compared to the negative controls. The negative controls showed no difference between them, despite the different nanoparticle composition and physicochemical properties. Binding in the negative control was attributed to exposed NeutrAvidin and defects, which we identified as uncured PDMS oligomers that leached and settled onto the substrate as aggregates.

Extraction of these uncured oligomers via serial solvent exchanges of the PDMS prior to glass bonding reduced the overall incidence of defects (Supplementary Fig. 3)[45]. As a second validation step, we performed an in-chip dose-response assay by assigning each sensing channel to a different concentration in the range of $1.6–28.1 \times 10^{10}$ NPs/mL (Fig. 4D). The number of localisations showed a linear dependence ($R^2 = 0.996$) up to a concentration of $7.1 \times 10^{10}$ NPs/mL corresponding to 200 counts per 100 µm².

The retrieved particle densities from the dose-response assay defined the upper and lower limits of detection of the platform. Although the optical system had single particle sensitivity, the intrinsic substrate defect density and non-specific binding imposed a lower limit of detection higher than the optical sensitivity, the lowest on the order of 0.5 counts per 100 µm². This problem of false positives due to the lack of signal specificity in the detected scattered signals is common to all label-free approaches based on elastic scattering. This is because any particle with a different refractive index than the surrounding media will elastically scatter light and thus contribute to a false positive detection signature. Additional imaging processing can eliminate contributions from intrinsic substrate defects but not from the non-specific bindings. For instance, one could obtain reference image scans of the same area prior to the addition of the analyte of interest and mask out all localisations that were already present in the sample in a routine, analogous to differential-based imaging but with an added step of image registration and alignment. Alternatively, one could switch to a conventional differential imaging approach, i.e. without scanning the FOV across the sample, at the expense of decreasing the throughput.

Regarding the upper detection limit, the likelihood of encountering more than one particle per diffraction limit imposes a boundary

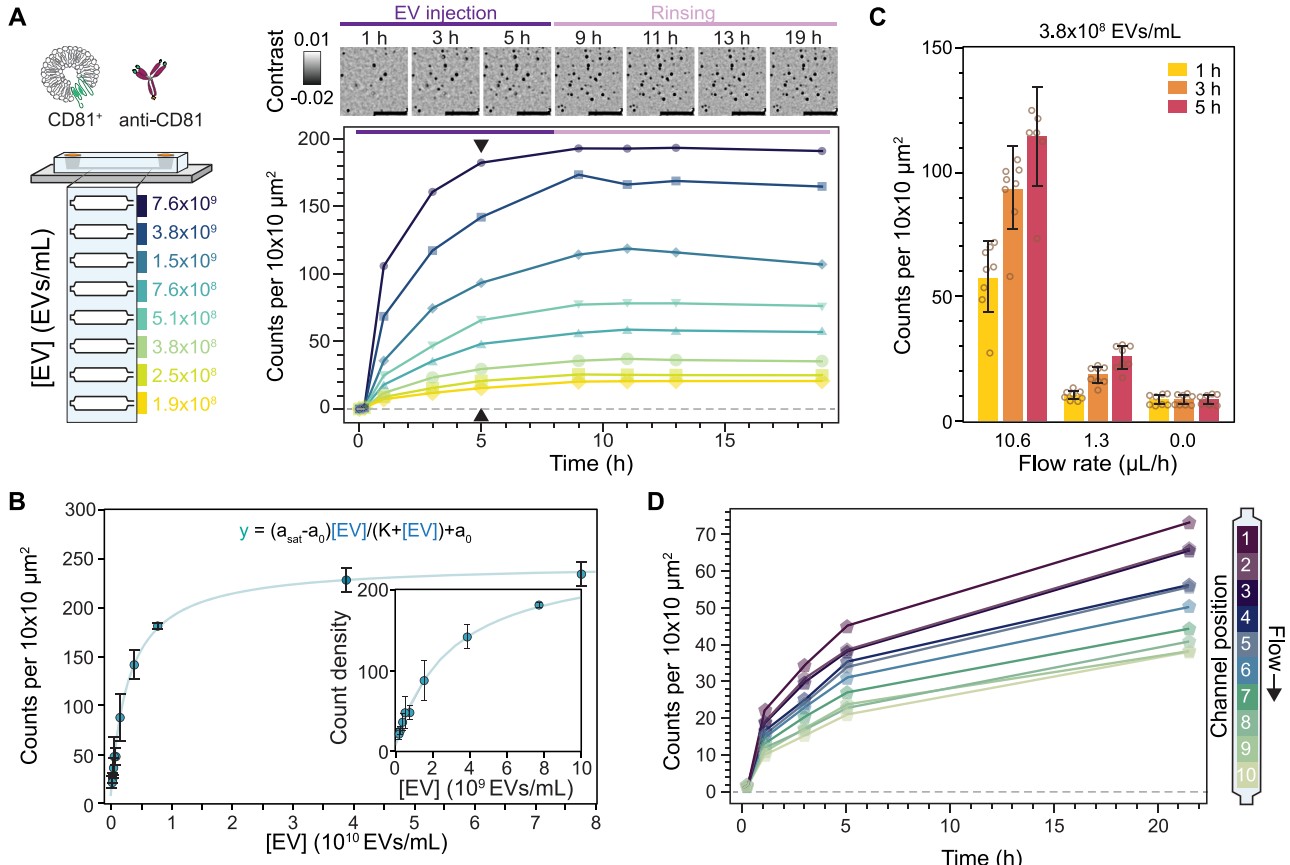

**Fig. 5 | In-chip EV binding kinetics. A** In-chip dose-response assay for CD81+ TiOSE4 EVs with each sensing channel loaded with a different EV concentration. Different colours encode each EV concentration. Top: representative zoom-in time-lapse images showing the binding kinetics upon EV injection and subsequent buffer rinsing. Scale bars: 5 µm. Bottom: measured binding kinetics expressed in terms of the number of captured EVs. Each data point corresponds to the mean of scans covering an area of 0.2 mm². Black arrows indicate the time-point considered as steady-state. **B** Corresponding dose response at steady-state with solid line representing a Langmuir model fit. Data points represent mean ± SD over independent channel area scan replicates (N = 3). **C** Effect of flow rate on the binding kinetics at a fixed EV concentration. Data were expressed as mean ± SD over independent channel scans covering an area of 0.2 mm² (N = 8). **D** Spatially resolved intra-channel dose-response kinetics under mass transport limited conditions. At flow rates below 1.3 µL/h, EV sample concentration gradients develop. Each data point corresponds to the mean of a 0.02 mm² segment of the total scanned area (1/10th) as indicated in the diagram to the left. The arrow along the diagram indicates the direction of flow, making channel position 1 the entrance of the sensing region.

to the detectable particle concentration, which for our optical system occurs at values above 200 localisations per 100 µm² (2 per µm²). Although by diffraction-limited density considerations alone, the upper value would correspond to about 16 localisations per µm² for a lateral resolution on the order of 250 nm; the experimental value is 8× lower due to a combination of factors: (i) the difficulty of packing particles into a dense monolayer due to electrostatic interactions and steric hindrances, (ii) the working principle of the single particle localisation algorithm and (iii) the fact that this algorithm operates on single images. Regarding the last two points, the algorithm relies on differentiating between foreground and background pixels to construct an SNR-based threshold—a task that becomes increasingly challenging at higher particle densities. As possible alternatives to extend this upper boundary, one could perform time-lapse differential imaging, akin to super-resolution-based imaging; or simply tune the biotin doping ratio (Supplementary Fig. 4). The latter would allow the sensor to operate in more physiological conditions for studying protein-protein interactions (i.e. µM and mM)[46].

### Immunoassay applied to EV samples

To demonstrate the compatibility of the sensing platform with complex heterogenous biological nanoparticle systems that to date remain challenging to characterise, we used EVs as an ideal model system. In detail, we pulled down CD81+ EVs derived from the ovarian cell line

TiOSE4, due to their higher tetraspanin expression levels[47], and studied the resulting binding kinetics as a function of EV concentration and flow rate.

Figure 5A illustrates an in-chip dose-response assay where EVs were continuously introduced at a flow rate of 10.6 µl/h for 8 h followed by PBS buffer rinsing at a flow rate of 1.3 µL/h for 11 h. Here the concentration was varied between $1.9 \times 10^8$ and $7.6 \times 10^9$ EVs/mL, as determined by NTA, with a different concentration assigned to each sensing channel. In this assay, we observed the number of captured CD81+ EVs plateauing between 5 and 9 h, indicating steady-state conditions. As expected, the associated binding rate and steady-state particle density depended on the EV concentration. In contrast, we were unable to retrieve a reliable dissociation rate constant, $k_{off}$, as EV unbinding events were barely detected during the buffer exchange (Fig. 5A inset). Figure 5B shows the results of three replicates of the in-chip assay together with two additional assays at much higher concentrations which were then fit to a standard Langmuir model. These data showed the same upper limit of detection as in Fig. 4D. The higher degree of variability in the dose-response averaged across three replicates stemmed from slight chip-to-chip variations in the form of defects along the channels which caused changes in the effective flow rate. These changes in effective flow rate manifest as discrepancies in the dose-response curves within a chip for two different concentrations, visible in Fig. 5A for the $5.1 \times 10^8$ and $7.6 \times 10^9$ EVs/mL curves.

Nevertheless, averaging over several chip replicates minimises these fabrication-based artifacts. Furthermore, the intra-chip results show that the sensor can quantitatively determine the relative abundance of EVs expressing a certain target molecule, thus enabling the fingerprinting of EV populations from a panel of surface protein biomarkers.

Overall, the measured binding kinetics are much slower compared to reaction-limited single antibody–antigen interactions and similar surface-based immunoassays[47–49]. We can explain this through a combination of three factors: mass transport limited reactions, EV avidity and sensor attainability. Firstly, the mass-transport limit effectively lowers the association rate, $k_{on}$, expected from reaction-limited kinetics[50]. Secondly, the EV avidity reduces $k_{off}$ due to the possibility of multivalent interactions, as a single EV can express the same target protein[51]. Thirdly, the high surface coverage of capture antibodies, attinebility, also effectively reduces $k_{off}$ thanks to the increased EV reattachment probability upon unbinding provided by proximal capture antibodies[51].

The combination of these three factors leads to complex binding kinetics, which some groups have modelled by introducing additional slow and fast rates for both $k_{on}$ and $k_{off}$[48]. Nevertheless, retrieving reliable rate constants for sensing systems where one of the binding partners is always in great excess relative to the expected dissociation constant, $K_D$, is prone to significant biases. Under these conditions, known as the titration regime, the equilibrium favours the formation of antibody–antigen complexes, and $K_D$ no longer reflects the concentration upon which half the binding sites are occupied[52]. We confirm our system falls within the titration regime by considering that one of the binding partners, the captured antibodies, are approximately three orders of magnitude higher than the typical antibody–antigen $K_D$ values ranging between 0.1–10 nM. We specifically computed the capture antibodies to be in the µM range given an estimated density of 0.08 pmol/cm² and microfluidic channel height of 10 µm. To focus the scope on fingerprinting, we restricted the analysis of the binding kinetics to solely determine the time to reach steady-state dynamics and the respective binding densities.

To confirm whether our system is reaction or transport-limited, we computed the ratio of reaction to mass transport rates characterised by the Dahmköhler number (Da), which was on the order of Da ~1–10, with values above one indicating mass transport limited kinetics[53]. In our case, the low analyte flow rates, the high capture probe density of our system, and the lower diffusion coefficient of the EVs relative to typical immunoassays, i.e. 3 µm²/s (EVs) vs 70 µm²/s (proteins) push the system into the mass transport limited regime. For mass-transport limited reactions, the flow rates can be further exploited to tune the binding kinetics of the system[54], as shown in Fig. 5C. Namely, increasing the flow rate concomitantly increases the number of captured EVs and pushes the system towards reaction-limited kinetics; albeit at the expense of low sample utilisation (capture efficiency). For example, after 5-h EV incubation, we observed three-fold (1.3 µL/h flow rate) and 14-fold (10.6 µL/h) improvements in EV capture over no flow conditions. Conversely, decreasing the flow rate exacerbates the mass transport limited binding kinetics, thereby increasing the sample utilisation (capture efficiency), which in turn leads to analyte concentration gradients along the sensing channel. Nevertheless, because our platform is based on recording large fields, and keeping this spatial information, these concentration gradients can be exploited for in-channel dose-response experiments, as shown in Fig. 5D.

With knowledge of how EV concentration and flow rate affect the binding kinetics, we designed our immunoassay to operate at low flow rates yet with minimal volumes of high sample concentrations. On the one hand, the low flow rates maximise the capture efficiency but have lower overall EV binding densities; while on the other hand, the high EV concentrations (in the range of $10^{10}$ EVs/mL) compensate for the expected lower binding densities, slower kinetics, and allow for the detection of low expression biomarkers. That said, the platform could target lower EV sample concentrations by increasing the flow rates.

## Molecularly fingerprinting EVs from ovarian cells

To validate that our optofluidic platform is suited to generate unique molecular fingerprint heterogeneous nanoparticle populations, we tested our system with four different ovarian cell line-derived EVs. Of these ovarian cell lines, three are cancerous (CaOV3, OV90, and ES2) and one benign (TiOSE4). In detail, we profiled these EV populations using a panel of six surface biomarkers and a negative control. We designed a microfluidic chip that integrated all functionalisation steps into a single device, i.e., bilayer formation, immunoassay assembly, and EV immunocapture (Fig. 6A). For molecular profiling, spatially separated sensing channels were independently functionalised with the following antibodies: IgG1 as negative isotype control; anti-CD9, anti-CD63, and anti-CD81, as three classical tetraspanin markers; anti-CD326 (EpCAM), anti-HE4 and anti-CA125, as three ovarian cancer biomarkers. Although not typically associated with EV profiling, both CA125 and HE4 are routinely used in clinical settings to detect ovarian cancer in blood[55].

We ensured robust statistics in these assays by imaging areas of 0.2 mm² per channel, resulting in more than $10^4$ detected vesicles for each biomarker. Figure 6B shows the EV binding kinetics for CaOV3 EVs as a function of positively expressed biomarker for a single microfluidic chip, indicating that the steady state is reached after 5 h. To generate the unique fingerprints, we computed the total EV counts at steady state over two experimental replicas (Fig. 6C). For the negative control, IgG1, we observed a binding density two- to three-fold higher relative to the baseline defects—indicating a small degree of non-specific binding, which is expected upon working at these higher EV concentrations. Comparison with conventional BSA passivation showed that SLBs minimised non-specific binding on average 4.5-fold better (Supplementary Fig. 5). Nevertheless, for all surface biomarkers we detected signals above the negative control.

To determine whether the fingerprints are unique enough to differentiate amongst different EV populations, we repeated this measurement with a minimum of three chip replicas for all EV samples. To correct for different levels of non-specific binding, the average count density of the negative control was subtracted from each surface marker on a chip-by-chip basis. Then to correct for differences in EV concentration and variations in flow rate between chips and EV samples, each fingerprint was normalised to the average expression level of the three tetraspanin markers (Fig. 6D). We opted for such normalisation to determine which biomarkers were positively expressed above the level given by the non-specific binding and for the fingerprints to be independent of EV concentration and flow rate. Overall, the pan-EV tetraspanin markers showed higher expression levels compared to other markers, in agreement with other recent works[3–5,47]; and displayed visible differences between the EVs populations, which have been shown to be sufficient to differentiate between EVs from different cell lines[7]. For the benign cell line TiOSE4 and the cancer ES2, only the pan-EV tetraspanins were positively detected[56], with the cancer biomarkers showing the same expression levels as the negative control. In contrast, EVs from the cancerous cell lines OV90 and CaOV3 showed positive expression levels for all cancer biomarkers to varying degrees; yet, both EV populations followed a general surface protein expression trend of CD326 > HE4 > CA125. Combining the pan-EV tetraspanins markers with the cancer-specific ones and performing a dimensionality reduction via principal component analysis (PCA) confirmed that the EV fingerprints from each ovarian cell line are unique and could be differentiated as indicated in the 2D projection (Fig. 6D). An exception occurred for one of the fingerprints of OV90, light purple point in PCA 2D projection, where a defect in the CD63 channel affected the overall flow and thus binding kinetics. This defect resulted in a

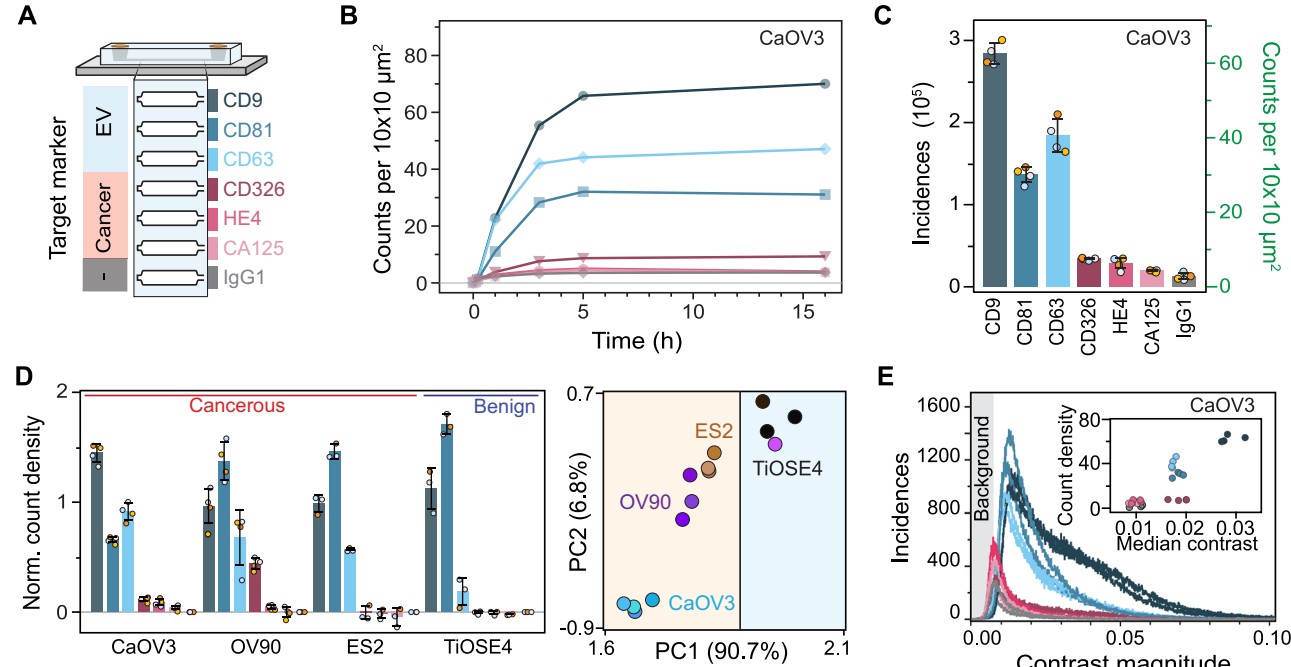

**Fig. 6 | Multiplexed EV fingerprinting. A** Schematic representation of the multiplexed in-chip immunoaffinity assay used to profile extracellular vesicles expressing different surface markers. Different colours encode the antibody used to target a specific surface marker. Each sensing channel was independently functionalised with a different capture antibody. **B** In-chip binding kinetics CaOV3 EVs expressing their respective surface protein markers. **C** Molecular fingerprint of CaOV3 EVs expressed in terms of the number of EVs captured within each channel after 5 h of continuous flow. The captured vesicles are given in both total and count density incidences. Data were expressed as mean ± SD over independent chip replicates (N = 4). **D** Normalised molecular fingerprint of four different ovarian cell line-derived EVs alongside the corresponding 2D projection after dimensionality reduction with principal component analysis (PCA). Data were expressed as mean ± SD over independent chip replicates (N = 4,4,3,3). Vertical line in the 2D projection separates EVs cell lines into cancerous (orange area) and benign (blue area) types. **E** Representative contrast distribution for all EVs captured within a single CaOV3 EV fingerprinting assay. The shaded area indicates the background contrast region that is not considered in the EV count density metric. Inset: correlative scatterplot comparing the median contrast magnitude against the count density of each marker for independent chip replicates of CaOV3 EVs (N = 4).

much lower expression level of CD63 compared to the other three chip replicates. Furthermore, such dimensionality reduction also differentiated between EVs originating from benign and cancerous cell lines as indicated by the vertical line separating the two different shaded regions; thus, demonstrating the potential of the platform to characterise different subpopulations expressing specific target analytes within a highly heterogenous nanoparticle sample.

In addition to the total number of EVs expressing a specific surface protein biomarker per channel, our platform also provided insight into the heterogeneity within, and amongst different EV distributions. Namely, Fig. 6E shows the distinct contrast distribution from the EVs captured in each channel. To recall, the magnitude of the signal contrast depends on the effective refractive index and size of the particles, which makes absolute sizing non-trivial and susceptible to errors when neither of these quantities is known a priori—even considering calibrating the system with reference materials. Nevertheless, if we assume that the refractive index of the different sized EVs is similar so that the measured contrast variations are mostly influenced by the EV size, then Fig. 6E inset delivers qualitative information about the relative EV-size differences. For instance, upon plotting the average contrast against the binding densities for the different biomarkers, we observed a positive correlation between the contrast of individual EVs and the total binding densities; namely, higher contrast EVs, associated with larger sizes, are more likely to be captured on the surface. We can hypothesise that this correlation is linked to a higher probability of interactions of larger EVs with proximal capture antibodies (attinebility), and possibly also higher number of target surface proteins per EV (avidity).

## Limitations

In this work, we aimed to develop a platform that reproducibly extracts unique fingerprints with minimal sample volumes from different EV populations based on a panel of surface protein markers. Namely, the platform was optimised to detect the relative surface protein expression levels of concentrated EV samples ($10^{10}$ EVs/mL). However, the system could be optimised to perform absolute EV quantitation or detection of rare EV biomarkers by either tuning the microfluidic chip geometry, flow rate, and/or the concentration of capture probes. For instance, to speed up the time response, the flow rates can be adjusted to match standard state-of-the-arts which are in the range of 1–10 μL/min. Similarly, the detection of lower concentrations can be accomplished by both increasing the channel height and flow rate to enter the reaction-limited regime.

Regarding the type of sample, the system has been validated with cell line-derived EVs in proof-of-concept experiments. Nevertheless, from a potential diagnostic perspective, the substrate passivation must be improved before working with clinically relevant EV samples, such as blood or plasma-derived EVs. Prior work on non-fluid solid-supported lipid bilayers with added PEGylation steps[41] or fast single-step PEGylation with PDMS-compatible solvents[57] offer promising routes to improve surface passivation. In addition, the high versatility of the lipid bilayer system offers multiple routes for minimising non-specific binding, and thus improving the overall performance of the platform, such as: (i) tuning the lipid composition, (ii) modifying the attachment chemistry of the capture probe from neutravidin-biotin linkers to direct covalent interactions and (iii) varying the capture probe from antibodies to aptamers[58,59].

In this work, we used a small selection of EV samples and experimental replicas as a proof-of-concept experiment to generate unique fingerprints. Nevertheless, to extend this work from mere EV population differentiation into a robust classification library[60], further validations are required. Specifically, from a statistical perspective, there is a need to increase the number of experimental replicates and the number of different EV samples, corresponding both to technical and biological replicates. Similarly, the number of required biomarkers in the fingerprint to generate a robust classification should be further investigated.

## Outlook

This work describes a route for multiplexing and profiling biological nanoparticles in solution based on spatially separated channels on a microfluidic chip. Microfluidic designs presented here could be readily extended by increasing the number of independent channels common to more complex microfluidic devices[61], with the size and complexity of the device becoming the ultimate limits in terms of multiplexing. Nevertheless, as a complimentary route, our platform is also fully compatible with single-molecule fluorescence read-out approaches[62], and thus could be combined with state-of-the-art fluorescently-tagged antibody[3,4,63,64] or aptamer[58,65] libraries to enable large-scale single particle profiling. We envision that our platform, when combined with on-chip standard additions approaches[66] and improved surface passivation could enable diagnostic and care monitoring of diseases based on a selection of disease biomarkers[60].

In summary, we show that our optofluidic platform integrates necessary assay steps to molecularly profile a population of heterogeneous biological nanoparticles such as EVs in a label-free manner, with single particle sensitivity, robust statistics, and a high degree of reproducibility. We demonstrated that our optical read-out allows us to in situ monitor the progress of each step of the assay, and thus optimise the surface functionalisation protocol in terms of robustness, sample preparation time, and high quality of the resulting coatings. We further highlighted the capabilities of our approach to study the underlying heterogeneity of EVs by combining information from the biomarker population as well as its contrast information. We foresee, that upon decoupling the size and refractive index dependence on the particle contrast[27,43,67], these assays would pave the way for approaches that can better characterise and study the heterogeneity of EVs by combining molecular fingerprinting, with size and effective material composition information at the single EV level.

# Methods

## Reagents

Bovine serum albumin (BSA, A2934, Sigma-Aldrich) solutions were prepared in phosphate-buffered saline (PBS, pH 7.4, 806552, Sigma-Aldrich). Biotinylated neutravidin (10443985, Fisher Scientific) was diluted to a concentration of 0.02 mg/mL (0.3 μM) in 1% BSA. Biotinylated antibodies anti-CD63, anti-CD326(EpCAM), anti-CD9, and anti-CD81 from Ancell (215-030, 126-030, 156-030, 302-030); anti-CA125 and anti-HE4 from LSBio (LS-C86749-1, LS-C743705-50) and Mouse IgG1 k-isotype control from Biolegend (400-104) were prepared to a concentration of 0.05 mg/mL (0.33 μM) in 3% BSA for all experiments. A custom AH peptide with the following sequence SGSWLRDVWDWICTVLTDFKTWLQSKLDYKD was synthesised by Proteogenix. A stock solution of 1 mg/mL was prepared by dissolving the lyophilised peptide in Milli-Q water according to the manufacturer's recommendation. This stock solution was aliquoted and stored at −20 °C for up to 1 month. For all experiments, the peptide stock solution was diluted to 0.45 mg/mL (200 μM).

## Liposome preparation

All liposomes were composed of 1-palmitoyl-2-oleoyl-*sn*-glycero-3-phosphocholine (16:0-18:1 PC) (850457 C, Avanti Polar Lipids, Inc) doped with 1,2-dioleoyl-sn-glycero-3-phosphoethanolamine-*N*-(cap biotinyl) (18:1 Biotinyl Cap PE) (870273 C, Avanti Polar Lipids, Inc) in a 99:1 molar ratio. To prepare the liposomes, the two lipid stock solutions in chloroform were first mixed and subsequently dried with a nitrogen stream and then placed under vacuum for 24 h. The dried lipids were then rehydrated to a concentration of 5 mg/mL in TRIS Buffer (100 mM, pH 7.4, 648315, Sigma-Aldrich) and vortexed for 2 min. These solutions were stored in the freezer for up to 1 month. Liposomes were then prepared using two approaches: sonication and extrusion. For bath sonication, the 5 mg/mL lipid solution was sonicated for 20 min at room temperature. For extrusion, the hydrated lipid solutions were passed 21 times through polycarbonate membranes, ranging in size from 200 to 30 nm, using Avanti Polar Lipids Mini-Extruder (610000). To prepare the smaller liposomes, i.e. 50 and 30 nm, the vesicle suspensions were serially extruded through successively smaller membrane pore sizes. All experiments were performed using vesicle suspensions at 1 mg/mL. Finally, 5 μL of a 500 mM CaCl₂ solution in Milli-Q water (Millipore) was added to the 500 μL 1 mg/ml liposome dilution. Unless stated otherwise, all liposome solutions were used within 5 days of preparation to minimise ageing effects. For sizing, liposome preparations were diluted and subsequently measured in triplicate using a commercial DLS (Malvern Zetasizer Nanoseries Nano-ZS, $T = 25$ °C, 173 backscatter detection).

## Fabrication of microfluidic chips

Microfluidic chips (MF) were fabricated using two-layer soft lithography. Two moulds were made on silicon wafers using a laser writer (Heidelberg uMLA, 365 nm), one for the flow layer using AZ P4620 (Microchemicals, GmbH) and one for the control layer using SU8 1060 (Gersteltec). The MF chips are made from polydimethylsiloxane (PDMS, Sylgard 184) mixed at a ratio of 10:1 polymer to curing agent. To make the thinner flow layer, the PDMS was spin-coated onto the wafer resulting in a thickness of about 30 μm. For the thicker control layer, the PDMS was dropcast onto the control mould to achieve a thickness of about 5 mm. The PDMS was then degassed under vacuum for two hours before baking in a convection oven at 80 °C for 1 h. Once cured, the PDMS on the control mould was peeled from the wafer, and the resulting control chips were cut out, and holes were punched. The control layer chips, together with PDMS covered flow wafer were treated with oxygen plasma (Diener Electronic, Atto 13.56 MHz, 10.5 L) for 1 min (300 W, 1.5 sccm) before being aligned under a stereo microscope. To bind them together, the aligned chips were baked for one hour in an 80 °C oven. The bound chips were removed from the flow wafer and the holes in the flow layer were punched. To complete MF chip assembly, the resulting two-layer MF chips and cleaned glass coverslips (24 × 40 mm², 0.17 mm, Karl Hecht) were exposed to oxygen plasma for 1 min, bound together, and baked at 80 °C for 1 h.

## Fabrication of microwells

The microwells were made by punching 5 mm holes into unpatterned cured PDMS of the same thickness. These were bound to cleaned glass coverslips using the same process described for the MF chips.

## On-chip bilayer formation and immunoassay functionalisation

To begin the on-chip bilayer formation at room temperature (21 °C), reagents were loaded into medical-grade microfluidic tubing (AAD04103, Tygon) and connected to the MF chip. For the peptide specifically, the tubing was first primed with 3% BSA solution for 30 min to reduce non-specific binding. To begin, the chip was primed with PBS until all the air within the channels was removed. Then, the liposomes were flowed into the channels until the bilayer had visibly formed, ~1 min under our experimental conditions. Once bilayer formation had occurred, the channels were rinsed with PBS for 1 min to remove any excess unbound liposomes. After rinsing, the AH peptide solution (200 μM in milli-Q water, pH = 6.8) was continuously flowed

through the channel for ~1– 2 min, or until no further vesicle rupture was visible. The bilayer was then rinsed with PBS for 5 min to remove all the peptides. This results in a fully formed bilayer coating. Next NeutrAvidin was flowed into the channel for 3 min and incubated for 30 min. The channel was again rinsed with PBS for 5 min and then the chosen capture antibodies were flowed into the channels for 3 min and incubated for 30 min. A final rinsing step with PBS for another 5 min completed the immunoassay. For the fingerprinting assay, the target EV sample was flowed at a rate of 1.3 μL/h for at least 5 h. Specifically, the EV stock solutions were diluted in 0.1% BSA to a target concentration in the range of $2–4 \times 10^{10}$ EVs/mL on the day of the experiment. All functionalisation steps were further validated with paired positive and negative control assays (Supplementary Fig. 6).

## Microwell bilayer formation

To begin the bilayer formation, the microwells were primed with 20 μL of PBS. Then 20 μL of 1 mg/mL of liposomes were injected. Once bilayer formation had occurred, 20 μL of liquid was removed prior to washing. Washing involved adding 50 μL of PBS into the microwell, followed by pipetting up and down ten times and subsequently removing 50 μL of liquid. This process was repeated five times. Then 20 μL of 200 μM AH was added and mixed by pipetting up and down ten times. The peptide was left in the microwell between 10 s to 1 min depending on its efficiency in rupturing liposomes. Finally, an additional step of washing was performed, resulting in the final bilayer.

## EV isolation from cell lines

The human ovarian cancer cell lines, including CaOV3, OV90, and ES2, were purchased from the American Type Culture Collection (ATCC: HTB-75, CRL-3585, CRL-1978). The benign cell line, TiOSE4, was obtained from transfection of hTERT into NOSE cells maintained in 1:1 Media 199: MCDB 105 with gentamicin (25 μg/mL), 15% heat-inactivated serum, and G418 (500 μg/mL) (Clin. Cancer Res. 2015, 21, 4811–4818). CaOV3 and ES2 cell lines were cultured in DMEM (Hyclone) and McCoy's 5A (Gibco), respectively. In addition, OV90 and TiOSE4 cell lines were maintained in RPMI-1640 (Hyclone). All basal media were supplemented with 10% foetal bovine serum (FBS, Thermo Fisher Scientific), 100 U/mL penicillin, and 100 μg/mL streptomycin (Cellgro) at 37 °C in 5% $CO_2$. EVs were isolated as previously reported[68]. In brief, cells were cultured to 80–90% confluence in a basal medium and washed with PBS to remove unattached cells and debris. Next, the cells were incubated in a conditioned medium supplemented with 1% Exosome-depleted FBS (Thermo Fisher Scientific), 100 U/mL penicillin, and 100 μg/mL streptomycin for 48 h. The medium was collected and centrifuged with $300 \times g$ for 5 min at 4 °C to remove floating cells or large debris. The supernatant was passed through a 0.8 μm membrane filter (Millipore Sigma) and concentrated using a Centricon Plus-70 centrifugal filter (MWCO = 10 kDa, Millipore Sigma) with $3500 \times g$ for 30 min at 4 °C. The sample was then loaded onto the size-exclusion chromatography (SEC) column packed with 10 mL of CL-4B Sepharose (Cytiva). The fractions of 4 and 5 (a total of 2 mL) were collected and concentrated with the Amicon Ultra-2 Centrifugal Filter (MWCO = 10 kDa, Millipore Sigma). The 1x protease and phosphatase inhibitor was added and stored at −80 °C until use.

## EV characterisation

EVs were lysed in LIPA lysis buffer (Cell Signaling Technology) for western blot analysis to confirm the characteristic EV biomarkers (CD9, CD63, and CD81). The blots were probed with flowing primary antibodies: anti-CD9 (1:500 dilution, BD Biosciences, 312102), anti-CD63 (1:500 dilution, Ancell, 215-820), and anti-CD81 (1:500 dilution, Santa Cruz Biotechnology,sc-166029). Chemiluminescence was detected using an Azure 280 imaging system (Azure Biosystems). The concentrations and sizes of EVs were determined by nanoparticle tracking analysis using Nanosight NS300 and were found to be

$1.22 \times 10^{11}$ particles/mL (CaOV3), $2.7 \times 10^{11}$ particles/mL (OV90), $3.0 \times 10^{11}$ particles/mL (ES2), and $3.8 \times 10^{11}$ particles/mL (TiOSE4).

## Microscope

The custom-built spatially incoherent digital holographic optical system was based on a common-path microscope operating in reflection, whereby illumination and imaging arms were separated by a single 50:50 beamsplitter plate (BSW27, Thorlabs) and all optics were arranged in a $4f$ configuration. In the illumination arm, a 455 nm light emitting diode (M455F3 LED, Thorlabs) was coupled into a 200-μm multimode fibre (M25L02, Thorlabs). Light outcoupled from the fibre using a 6.24 mm aspheric lens (A110TM-A, Thorlabs) was then relay imaged onto the sample plane formed by a 1.46 NA oil immersion objective (APON 60XOTRIF, Olympus) via a 1:1 imaging system, composed of two 300 mm achromatic doublet lenses (AC508-300A, Thorlabs). Under this optical arrangement, the NA of illumination was approximately 0.5, resulting in a flat-top illumination with a diameter of 89.5 μm. For the imaging arm, light collected from the sample by the objective and reflected off the 50:50 beamsplitter was imaged onto a scientific CMOS camera (C11440-22CU, 6.5 μm pixels, Hamamatsu) using a 300 mm achromatic doublet (AC508-300A, Thorlabs) resulting in a 100× magnification. The sample was mounted on a motorised XY microstage (Mad City Labs) equipped with linear encoders, as well as an XYZ nanostage (Nano-LP200, Mad City Labs). The sample focus position was stabilised to within 10 nm using the backreflection from a 670 nm misaligned confocal beam with a low numerical aperture of illumination (CPS670F, Thorlabs). Specifically, the beam position was used as a feedback parameter in the proportional–integral–derivative loop.

## Optical imaging

For all experiments, we measured power at the sample between 1.4–1.7 mW equivalent to an irradiance of 0.22–0.27 μW/μm². During acquisition, a field of view of 66.6 μm × 66.6 μm corresponding to an area of 1024 × 1024 camera pixels was recorded with an exposure time of 10 ms and a fixed frame rate of 100 Hz. To minimise data load and increase the signal-to-noise ratio, the data were saved in the form of 100 time-averaged frames, leading to an effective time resolution of 1 Hz. Prior to each data acquisition, an experimental flat-field image was generated and saved. The flat field imaged was produced by first collecting a stack of at least 60 time-averaged frames taken at different sample locations and the same focus position, and subsequently taking the median value on a pixel-by-pixel value. This flat-field image contained inhomogeneities along the optical system and imperfections in the sample illumination.

## Image processing

All images were first normalised to the average background camera counts in the background. Next, we flat field corrected the normalised images, by division, to remove inhomogeneities attributed to the optical system and sample illumination. For image scans, the image stacks were stitched together using a phase correlation algorithm. To remove large feature contributions from the flat-field corrected images, such as out-of-focus objects corresponding to the top surface of the PDMS microfluidic device, a spatial median filter with a kernel size of 17 pixels was determined and subtracted. This process had no effect on the contrast or shape of the diffraction-limited spots.

## Particle localisation

First a global noise level from each image was estimated from the median absolute deviation. Next, a local noise estimate of each image was determined by computing the root-mean-square of all pixel values within a kernel size of 65 pixels falling within 2.5× the global noise estimate. This local noise estimate was then used to determine a signal-to-noise ratio image; i.e. by dividing the initial processed image by the

estimated local noise. Then, candidate regions of interest were segmented based on the following two selection criteria: (a) pixel-based: positive for all pixels exceeding a signal to noise threshold of 4; and (2) clustering-based: positive if there were a minimum of three pixels exceeding the SNR threshold within a $3 \times 3$ pixel[2] area. Diffraction-limited spots satisfying the selection criteria were then segmented and localised with sub-pixel precision using the radial symmetry centres algorithm[69]. The resulting lateral position, signal contrast and the integrated signal contrast were stored for further processing. The overall computational time for particle localisation per FOV ranged from 10 to 100 ms.

## Statistics and reproducibility

All imaging experiments are based on single particle detection read-out, which is governed by Poisson statistics. To obtain robust statistics of biological nanoparticles, we sampled a minimum of 10,000 particles per fingerprinting and kinetic experiment. We obtain this minimum number of particles by controlling the density of capture sites and the total sample area imaged. For chip reproducibility, a minimum of three replicates were performed. In all cases, the number of technical replicates are stated in the corresponding figure captions. Chips showing defects in fabrication or unsuccessful surface functionalisation were discarded from further downstream analysis. Unsuccessful surface functionalisation was determined by in situ imaging and characterising the number of defects within the imaging chamber. Representative data are shown for each experiment. All measurements were, by default, performed on random samples of biological nanoparticles that are loaded into the microfluidic tubing. The solution loaded into the tubing was randomly selected from an Eppendorf. No further experiment randomisation was performed. All data were analysed by the same software using the same parameters, ensuing that the analysis is blind to the type of sample. No further experiment blinding was performed.

## Reporting summary

Further information on research design is available in the Nature Portfolio Reporting Summary linked to this article.

# Data availability

The main data supporting the results of this study are available within the paper and its Supplementary Information. The raw and analysed datasets generated during the study are too large to be publicly shared yet are available for research purposes from the corresponding authors upon request. Requests will be fulfilled within 10 weeks. Source data are provided with this paper.

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

## Acknowledgements

The authors acknowledge the following funding sources: Swiss National Science Foundation grant 207485 (P.R., J.O.A., R.Q.); US National Institutes of Health grant R01CA229777 (H.L.); US National Institutes of Health grant U01CA284982 (H.L.); US National Institutes of Health grant R01CA239078 (H.L.); US National Institutes of Health grant R01CA237500 (H.L.); US National Institutes of Health grant R01CA264363 (H.L.); US National Institutes of Health grant R21CA267222 (H.L.); US National Institutes of Health grant U01CA279858 (H.L.); US National Institutes of Health grant R21CA217662 (H.I.); US National Institutes of Health grant R01GM138778 (H.I.).

## Author contributions

H.I., H.L., J.O.A. and R.Q. conceptualised the project. Method development was performed by A.S., J.G.G. and J.O.A. A.S., J.S.H., P.R., H.I. and J.O.A. performed experiments. J.O.A. developed the software. Formal analysis was conducted by A.S., J.S.H. and J.O.A. A.S. and J.O.A. were responsible for data visualisation. Supervision and funding acquisition was provided by H.I., H.L., J.O.A. and R.Q. The original draft of the manuscript was written by J.O.A., and subsequent review and editing were conducted by A.S., J.G.G., J.S.H., P.R., H.I., H.L., J.O.A. and R.Q.

## Competing interests

The authors declare no competing interests.
