## [Peer Review File · Nature Communications]

Reviewers' Comments:

Reviewer #1:

Remarks to the Author:

The authors present an optofluidic platform that brings together state-of-the-art digital holography with PDMS microfluidics by using SLB as a surface chemistry building block to integrate both technologies. Specifically, the authors thoroughly characterize the robustness and performance of the platform to satisfy three key parameters, in terms of sensitivity, high throughput, and molecular fingerprinting. The sensing principle and the functionalization procedure delineated in this study exhibit a novel approach. Nevertheless, there are certain aspects of the results that appear to require further clarification.

1. please describe the buffer contents when a fusogenic peptide causes osmotic shock.
2. in Fig. 2D, what is the difference between light grey and dark grey graphs?
3. In Fig. 2E, the authors described why AH-peptide preferentially ruptures liposomes with diameters below 125-150. Whereas, please indicate, including references, why the particle localization density of signal data was low at 200 nm-sized liposomes.
4. Please give details on conditions performing peptide incubation (including temperature and pH).
5. With Fig S2, why did the frequency of spontaneous rupture decrease when the liposome sample aged? please describe the reason including references.
6. In Fig S2, there is a noticeable difference between AH peptide data with liposome aging. Please show a bar graph of the count data and statistical analysis.
7. Please correct the "P" value for Chip 2 on page 9 in italics.
8. In Fig 4c, please indicate why 20 nm AuNP was chosen as the positive control. There is a difference in the density between AuNP and exosome (or liposome).
9. Please show the validation data of the immunoaffinity functionalization using bare AuNPs.
10. Please show the validation data of the immunoaffinity functionalization using a 30 nm-sized liposome functionalized with streptavidin.
11. Please give a detailed description of how elastic scattering causes the problem described in the second paragraph on page 10.
12. In Fig 4D, please show the coefficient of determination value in the linear range of the regression line.
13. In Fig 5A, the dose-response in concentration at 5.1 is higher than the concentration at 7.6 whereas, in Fig 5B, the dose-response decreases sequentially with concentration. Please explain why there is a mismatch.
14. In Fig 5B, please comment on the fitting model as you applied and show the coefficient of determination value in the linear range of the regression line.
15. In the sentence " To recall, the magnitude of the signal contrast is a function of both the effective refractive index and size of the particles, which makes absolute sizing non-trivial and susceptible with errors" the preposition with is inappropriate. Please change accordingly.

Reviewer #2:

Remarks to the Author:

The manuscript NCOMMS-23-45101-T titled: "Molecular fingerprinting of biological nanoparticles with a label-free optofluidic platform" by Alexia Stollmann and co-authors presents the proof-of-concept of a technology that integrates label-free imaging, microfluidics and surface chemistry at the aim to detect biological nanoparticles. The surface functionalization protocol is based on supported lipid bilayers (SLBs) and the label-free imaging is an in-line digital holographic microscope for the detection of nanoparticle at the limit of diffraction. The work is mainly organized in four parts: (i) Integration of surface functionalization and imaging procedure by defining image processing steps and thresholding to distinguish between background and signal, (ii) Assessments of microfluidic chip reproducibility and robustness (iii) validation of the immunoassay by flowing a sample of streptavidin-labelled 20 nm gold nanoparticles (AuNPs-SAv) and (iv) test of the technology by proving the biosensing performances on six surface biomarkers from 4 different ovarian cancer cell line.

The paper is technically correct and well-written. The data analysis support claims and conclusions and the authors clearly evaluate the limitations of the methodology. The paper is suitable for publication after minor review points will be addressed, specifically:

- In the introduction, the in-line holographic technique is not mentioned. In order to give a complete overview of the proposed method authors should declare it in this introductory section.
- Concerning the data presented in Fig. 2B, it's not clear how these results are obtained. Authors should add more information on this point.
- In the inset in Figure 2D concerning the background contrast there are different areas, what they refer to ? please explain better in the text or in the caption.
- Authors should add some information about the computational time of the image processing steps, i.e the time for detecting the spots in a single patch of 100x100 μm^2 and the total computational time due to the stitching of all the patches. In the supporting videos authors declare "real-time" imaging, what they refer to?
- Some sections may be improved in terms of readability. For example in the introduction the concept of coexistence of sensitivity, high-throughput, and molecular fingerprinting is repeated few times; a further example is at page 4 where the description of the imaging setup contains two times the FOV. I suggest to review the manuscript addressing these repetitions.

Reviewer #3:

Remarks to the Author:

In the manuscript by Stollmann et al., the authors describe a label-free optofluidic-based approach for fingerprinting biological nanoparticles (i.e., extracellular vesicles) at the single-vesicle level. Specifically, the authors designed a microfluidic platform to capture nanoparticles and nanometric liposomes by implementing the same concepts of an immunoaffinity assay. To bind the nanoparticles to the microchannels, a supported lipid bilayer (SLB) was first formed through rupture and fusion of liposomes at the surface of the microchannel and, then, antibodies were attached to the SLB using NeutrAvidin and biotin. The authors demonstrated that the binding of streptavidin functionalized nanoparticles can be detected at the single particle level and is affected by the particles concentration and flow rate of the dispersion within the microchannel. Finally, the authors demonstrated that the binding profile (i.e., fingerprint) of different ovarian cell line-derived EVs can be measured by using microchannels with different antibodies attached to their surface.

General comments:

In general, the paper by Stollmann et al. nicely describes the development and usage of a label-free optofluidic-based platform for detecting different types of EVs at the single vesicle level. Nevertheless, since this work is a proof of concept study, it mainly focuses on the development and testing of the technique rather than addressing various important challenges in EV detection (e.g., differentiating specific EVs from a sample that contains a mixture of EVs, detection of EVs from real biological samples, etc.). In terms of the experiments described in the paper, it is expected that surface characterization, such as SLB formation and antibodies attachment, will be assessed using other techniques like atomic force microscopy, fluorescence microscopy, etc. Similarly, quantification of binding/dissociation constant and evaluation of various parameters that may influence nanoparticles binding were not reported. This information is critical for determining the optimal conditions (flow rate, nanoparticle concentrations, solution pH, salt concentration, etc.) for nanoparticles binding. Finally, while profiling of EVs using different surface biomarkers was demonstrated as a proof-of-concept, the obtained fingerprints were not significantly different to enable clear differentiation between EV subpopulations. For these reasons, at this stage the paper by Stollmann et al. does not meet the criteria for publication in Nature Comm.

Additional comments:

1. Page 7: According to the authors, the formation of defect free SLB is critical for minimizing false positive readouts. Nevertheless, the formation of an intact SLB without defects/boundaries over a large surface area is improbable. Have you characterized the quality of the formed SLB using complementary methods like AFM, fluorescence microscopy, etc.?
2. Page 11 and figure 5A: Why the assay takes so long (hours)? Have you tested different flow rates except from the ones you reported? What about the density of antibodies at the SLB surface?

the paper reports a 99:1 molar ratio, why not higher than this? Are the antibodies/proteins washed away at higher flow rates?

3. Page 11: are the testes EV concentrations comparable to biological samples?

4. In page 11 you mention "...to retrieve a reliable dissociation rate constant...". I could find any estimation of the dissociation rate constant in the paper.

5. Figure 5A (time lapse images): time should be indicated in the image.

6. Figure 6: A major challenge in this area is developing methodologies that can differentiate the different types of EVs in a given biological sample. The work presented by the authors does not show this kind of ability. Can the method presented by the authors be utilized for detecting various EV subpopulations in a given biological sample?

7. Figure 6D: The counts for the ovarian cancer biomarkers is very low relative to the classical biomarkers which appear for all types of EVs. To differentiate between the different populations of EVs a significantly different binding profile is expected to appear for the cancer biomarkers.

REPLY TO REVIEWERS: NCOMMS-23-45101-T

REVIEWER

COMMENTS

Reviewer #1 (Remarks to the Author):

The authors present an optofluidic platform that brings together state-of-the-art digital holography with PDMS microfluidics by using SLB as a surface chemistry building block to integrate both technologies. Specifically, the authors thoroughly characterize the robustness and performance of the platform to satisfy three key parameters, in terms of sensitivity, high throughput, and molecular fingerprinting. The sensing principle and the functionalization procedure delineated in this study exhibit a novel approach. Nevertheless, there are certain aspects of the results that appear to require further clarification.

We thank the reviewer for taking the time to evaluate our manuscript and providing constructive feedback to improve our work. Below, we address the concerns point-by-point.

1. please describe the buffer contents when a fusogenic peptide causes osmotic shock.

When we refer to osmotic shock, we do not attribute this effect to the AH peptide, but rather to the difference in ionic strengths between the two aqueous solutions before and after incubation with the AH peptide. As stated in the methods, the AH peptide was only dissolved in MilliQ water, resulting in a low ionic strength solution. In contrast, all other aqueous solutions, including the rinse buffer, prepared in PBS (pH 7.4, 137 mM NaCl, 2.7 mM KCl, 10 mM Na₂HPO₄, 1.8 mM KH₂PO₄) were at a higher ionic strength. Introduction of the AH peptide therefore induces an initial hypotonic shock, as PBS is exchanged for the peptide solution in MilliQ water; followed by a hypertonic shock as the peptide solution is exchanged for PBS.

Action taken: We have added the following text in the main manuscript to clarify this point.

The osmotic shock is not due to the peptide, but rather by the differences in ionic strength between aqueous solutions prior and after incubation with the AH peptide.

2. in Fig. 2D, what is the difference between light grey and dark grey graphs?

Each gray graph in Fig. 2D represents the contrast distribution of all particle signatures identified over an estimated scanned substrate area of 0.2mm². These individual distributions, all having the same gray intensity, are then overlaid with a transparency level effect. Due to the number of experimental replicas being overlaid (N = 6), darker areas in the Fig. 2D simply reflect that there is a higher degree of overlap in that region. We have now clarified this and specified the number of replicas for this in the caption.

Action taken. We modified the caption of figure 2D to clarify the source of confusion.

(D) Particle contrast histograms from all localisations found in substrates exposed only to PBS solution (N = 6). Each curve with the same transparency level and grey intensity corresponds to an approximate scanned area of 0.2 mm². The overlap between different curves is indicated by the different degrees of transparency.

3. In Fig. 2E, the authors described why AH-peptide preferentially ruptures liposomes with diameters below 125-150. Whereas, please indicate, including references, why the particle localization density of signal data was low at 200 nm-sized liposomes.

The mechanism of support lipid bilayer formation is complex involving multiple competing pathways, each of which is dependent on multiple experimental parameters. Spontaneous liposome rupture is one such pathway and is strongly influenced by the substrate-liposome interactions – with the rate of spontaneous rupture increasing with vesicle size¹. Although the AH peptide preferentially ruptures liposomes smaller than 125-150 nm, the higher frequency of spontaneous rupture for the 200 nm liposome preparation population compensates for the lower fusogenic-induced rupture, ultimately leading to a lower defect density than the 100 nm preparation.

Action taken. We modified the text in the main manuscript discussing the Fig. 2E to:

We can rationalise these results by considering the interplay between two complementary size-dependent liposome rupture mechanisms: spontaneous rupture, with the rate of rupture increasing with size, and the peptide-induced rupture, which due to the membrane curvature sensitivity of the AH-peptide, preferentially ruptures liposomes with diameters below 125-150 nm. The size distribution from Fig. 2B confirms that the 30 nm liposome preparation has the smallest fraction of liposomes above the AH-peptide size cut-off, and thereby does not depend on the spontaneous rupture mechanism. As a corollary, the probability of having a higher proportion of unruptured liposomes unaffected by the AH peptide is significantly higher for all other preparations and now becomes dependent on the vesicle-substrate interactions, thereby leading to a greater variance.

4. Please give details on conditions performing peptide incubation (including temperature and pH).

Action taken. We added the details regarding peptide incubation into the methods section as follows:

On-chip bilayer formation and immunoassay functionalisation. To begin the on-chip bilayer formation at room temperature (21°C), reagents were loaded into medical grade microfluidic tubing (AAD04103, Tygon) and connected to the MF chip. For the peptide specifically, the tubing was first primed with 3% BSA solution for 30 minutes to reduce non-specific binding. To begin, the chip was primed with PBS until all the air within the channels was removed. Then the liposomes were flowed into the channels until the bilayer had visibly formed, approximately 1 minute under our experimental conditions. Once bilayer formation had occurred, the channels were rinsed with PBS for 1 minute to remove any excess unbound liposomes. After rinsing, the AH peptide solution (200 μM in milliQ water, pH = 6.8) was continuously flowed through the channel for approximately 1 – 2 minutes, or until no further vesicle rupture was visible. The bilayer was then rinsed with PBS for 5 minutes to remove all the peptide. This results in a fully formed bilayer coating. Next NeutrAvidin was flowed into the channel for 3 minutes and incubated for 30 minutes.

5. With Fig S2, why did the frequency of spontaneous rupture decrease when the liposome sample aged? please describe the reason including references.

Spontaneous liposome rupture is a complex self-assembly process that depends on multiple parameters from pH, substrate interactions, ionic strength, liposome composition and structure. Based on similar observations reported in literature using similar liposome system to form SLBs, the rate of spontaneous liposome rupture decreased with aging and was correlated with a decrease in liposome size^{2,3}. Specifically, Cho *et al* introduced a model based on a time-dependent changes in the liposome structure from larger ellipsoid particles to smaller spherical ones². In this model, liposome changes caused by structural relaxation reduce tension, lower the strength of liposome-substrate interactions, and thereby result in a higher energetic barrier for spontaneous rupture.

Action taken. We modified the text in the main manuscript discussing the Fig. S2 to:

As the liposome sample aged, we observed a decrease in the spontaneous rupture frequency alongside an increase in unruptured liposomes prior to peptide treatment (Fig. S2), in agreement with similar experimental work^{2,3}. In both these prior works, the decrease in rupture frequency associated with liposome ageing was correlated with a decrease in size of the liposome population. Besides a lower spontaneous rupture rate with decreasing liposome size, Cho *et al* proposed a model based on a time-dependent changes in the liposome structure to explain the higher amount of unruptured liposomes². In this model, structural relaxation of the liposomes from ellipsoidal to spherical-shaped particles lowers the interaction strength between the liposome and the substrate, resulting in a higher energetic barrier for spontaneous rupture.

6. In Fig S2, there is a noticeable difference between AH peptide data with liposome aging. Please show a bar graph of the count data and statistical analysis.

The data in Fig. 3B already contain the bar graph the reviewer is asking for. Specifically, the horizontal axis on the top of the bar chart explicitly assesses the quality of the lipid coating as a function of liposome ageing.

Action taken: We have incorporated the statistical analysis value into the main text and reference the Fig. 3B to avoid confusion as follows:

Nonetheless, upon peptide treatment there was no significant defect density dependence on liposome ageing (Fig. 3B, one way Anova: $P = 0.534$).

7. Please correct the "P" value for Chip 2 on page 9 in italics.

Action taken: Corrected, thank you for spotting this.

8. In Fig 4c, please indicate why 20 nm AuNP was chosen as the positive control. There is a difference in the density between AuNP and exosome (or liposome).

The principle behind the immunoaffinity pull-down assay is to target analytes located at the surface of a nanoparticle, irrespective of the nanoparticle internal composition and origin (i.e. biological or synthetic), and to immobilize them onto the substrate via antibody-antigen interactions. We chose the 20 nm streptavidin functionalized AuNPs as a positive control since their functionalization, surface density of streptavidin, size, and zeta potential have been characterized by the manufacturer and this translates into a homogeneous nanoparticle distribution with well-defined physicochemical properties. Furthermore, from an optical detection perspective, these AuNPs have been used as a model system to assess the performance of various interferometric-based label-free microscopes across different groups due to their high SNR and low coefficient of variation⁴⁻⁷. In short, the 20 nm AuNPs represent a homogeneous and well-defined nanoparticle population that is ideally suited to validate the immunoassay.

If the reviewer is referring to the surface density of target analytes, it is expected that there will be differences with respect to the surface density of a particular analyte on an extracellular vesicle -- an intrinsically unknown quantity a priori that one would like to quantify. Nonetheless, the streptavidin-functionalized 20 nm AuNPs represent the best-case scenario in terms of binding, due to the possibility of multiple antigen-antibody interactions, which in turned allowed us to quantify the upper detection limit of the platform -- hence another reason why this system was chosen as a positive control.

If the reviewer is referring to density in terms of mass per unit volume, again, we expect there to be differences between the two different particle types. Nevertheless since the measurements are performed in the low Reynolds numbers regime, where diffusion is the main driver behind particle motion and not sedimentation⁸, we argue that density plays no role in the immunoassay.

Action taken: We have included the following text in the main manuscript discussing the rationale behind the streptavidin 20 nm AuNPs as positive control.

The principle behind the immunoaffinity pull-down assay was to target analytes located at the surface of a nanoparticle, irrespective of the nanoparticle internal composition and origin (i.e. biological or synthetic), and to immobilize the nanoparticle onto the substrate via antibody-antigen interactions. We therefore validated the immunoassay by first flowing a sample of streptavidin-labelled 20 nm gold nanoparticles (AuNPs-SAv) as positive control, and bare 40 nm carboxylated gold particles and biotinylated-liposomes as negative controls against the antibody layer (Fig. 4C). The rationale for choosing AuNP-SAv as a positive control was that they represent a synthetically homogeneous nanoparticle population with well-defined physicochemical properties, which has been widely used to assess the performance of various interferometric-based label-free microscopes across different groups thanks to their high SNR and low coefficient of variation size and therefore particle contrast. As expected, the AuNPs-SAv showed nearly a 200-fold more binding compared to the negative controls. The negative

controls showed no difference between them, despite the different nanoparticle composition and physicochemical properties.

9. Please show the validation data of the immunoaffinity functionalization using bare AuNPs.

As per the reviewer's request, we have performed this additional experimental validation using a 40 nm carboxyl AuNPs.

Action taken: We have included the validation data into Fig. 4C and amended the text of manuscript and caption accordingly.

Fig. 4. On-chip immunoaffinity capture assay validation. (A) Representative zoom-in images of the substrate after each functionalisation step: bilayer formation, NeutrAvidin incubation, and biotinylated antibody incubation. Scale bars: 5 μm. (B) The number of localisations after each functionalisation step. (C) Validation of the immunoaffinity functionalisation using streptavidin functionalised AuNPs (AuNP-SAv) as a positive control, and carboxyl AuNPs and biotinylated liposomes as negative controls. Scale bars: 10 μm. (D) Dose response for different concentrations of streptavidin-functionalised 20 nm gold nanoparticles. Each data point corresponds to the mean of scans covering an area of 0.2 mm² over multiple different channels within each chip (N = 3). Error bars represent the standard deviation over the mean.

10. Please show the validation data of the immunoaffinity functionalization using a 30 nm-sized liposome functionalized with streptavidin.

We acknowledge the reviewer's request; however, we do not see the relevance of this assay considering the immunoaffinity pull-down assay is based on targeting specific markers expressed on the surface of a nanoparticle and not on the composition of the particle itself. Therefore, there is no difference whether the immunoassay is validated using streptavidin-

functionalized 20 nm AuNPs or 30 nm liposomes. As stated earlier, the advantage of using functionalized 20 nm AuNPs is that the surface coating, as well as the particle population, have been extensively characterized using orthogonal techniques as per the manufacturer's specifications. Furthermore, the signal contrast of 20 nm AuNPs is well characterized and narrowly distributed, which allows us to differentiate between more than one particle binding within the diffraction limited spot. Whereas the functionalized liposomes would require additional characterization and their particle population would be more heterogeneous as expected from the size distribution shown in Fig. 2B. In short, a validation assay with 30 nm-sized liposomes functionalized with streptavidin would only add experimental complexity without providing any new information than the assay with 20 nm sized AuNPs.

11. Please give a detailed description of how elastic scattering causes the problem described in the second paragraph on page 10.

Label-free platforms based on elastic scattering lack specificity to differentiate between signals arising from heterogeneous nanoparticle systems or from nanoparticles containing different surface markers. This stems from the fact that any particle, which is not the analyte of interest, with a different refractive index than the surrounding media will elastically scatter light and contribute to a false positive reading. In this work we introduce molecular specificity to our imaging system by using immunoaffinity pull down assays on supported lipid bilayers to target specific populations of analytes and to minimize non-specific binding. Nevertheless, any detectable scattering signal present prior to the immunoaffinity assay would be counted as false positives if no other image processing step than particle localization is considered. For our system, false positives originate from intrinsic defects in the glass substrate, non-specific binding of contaminants during any of functionalisation steps, defects in the SLB, unruptured vesicles, or even aggregation of antibodies. To remove these false positive contributions one must rely on temporal information, for instance by counting the defects present before the addition of the analytes and subsequently removing their contribution from the total density determination after analyte incubation.

Action taken: We have clarified the text in page 10 as follows:

The retrieved particle densities from the dose-response assay defined the upper and lower limits of detection of the platform. Although the optical system had single particle sensitivity, the intrinsic substrate defect density and non-specific binding imposed a lower limit of detection higher than the optical sensitivity, the lowest on the order of 0.5 counts per $100 \mu\text{m}^2$. This problem of false positives due to the lack of signal specificity in the detected scattered signals is common to all label-free approaches based on elastic scattering. This is because any particle with a different refractive index than the surrounding media will elastically scatter light and thus contribute to a false positive detection signature. Additional imaging processing can eliminate contributions from intrinsic substrate defects prior to the pull-down assay, but not from the non-specific bindings. For instance, one could obtain reference image scans of the same area prior to the addition of the analyte of interest, and mask out all localisations that were already present in the sample in a routine, analogous to differential-based imaging but with an added step of image registration and alignment. Alternatively, one could switch to a conventional differential imaging approach, i.e., without scanning the FOV across the sample, at the expense of decreasing the throughput.

12. In Fig 4D, please show the coefficient of determination value in the linear range of the regression line.

Action taken: We have added the coefficient of determination value directly into Fig. 4D ($R^2 = 0.996$) and modified the figure to highlight the portion of the data corresponding to the linear range.

13. In Fig 5A, the dose-response in concentration at 5.1 is higher than the concentration at 7.6 whereas, in Fig 5B, the dose-response decreases sequentially with concentration. Please explain why there is a mismatch.

Figure 5A shows the dose-response from a single chip replicate, whereas Fig. 5B corresponds to the dose-response from three independent chips. We argue that the discrepancy between the concentration 5.1 and 7.6×10^8 EVs/mL originates from small defects in fabrication along one of the channels, (7.6×10^8 EVs/mL), which causes differences in flow rate. This variation in flow rate, i.e. in the form of a decrease, leads to a lower number of captured EVs as quantified experimentally Fig. 5C. To compensate for these fabrication artifacts and variation of flow rates amongst the channels, we performed the dose-response assays in triplicate.

To support such hypothesis, we analysed instances of experiments of experimental replicates that showed large discrepancies within a single channel (Figure R1A). Upon inspection of the microfluidic chip with a big FOV imaging system, we identified that a large reduction in immobilised EVs was correlated with the presence of a defect in the channel, effectively altering the relative flow rate with respect to the other channels (Figure R1B).

Figure R1. Defects along the channels affect the immuno-affinity pull down assay. A) Schematic representation of an EV fingerprinting assay for three different chip replicates. Arrows indicate the biomarker where a large discrepancy between experimental replicates was observed. B) Channel inspection of chip 2 shows a large defect along the channel supplying EVs to the CD63 pull-down assay.

Action taken: We have included the following discussion in the main text as follows:

The higher degree of variability in the dose response averaged across three replicates stemmed from slight chip-to-chip variations in the form of defects along the channels which caused changes in the effective flow rate. These changes in effective flow rate manifest as discrepancies in the dose response curves within a chip for two different concentrations, visible in Fig. 5A for the 5.1×10^8 and 7.6×10^9 EVs/mL curves. Nevertheless, averaging over several chip replicates minimises these fabrication-based artifacts.

14. In Fig 5B, please comment on the fitting model as you applied and show the coefficient of determination value in the linear range of the regression line.

No model was fit in the original manuscript submission. We appreciate the reviewer’s suggestion and have fit a Langmuir dose response model for antibody-antigen interaction which results in the characteristic symmetric sigmoidal shape.

$$N_b(C) = N_0 + C(N_{sat} - N_0)/(K + C)$$

where N_0 , N_{sat} , K correspond to the number of bound particles in the absence of analyte, at saturation, and the concentration at half saturation, respectively. We have also linear fit the data in the respective regime and report the corresponding coefficient of determination ($R^2=0.963$).

Action taken: We have modified the Fig. 5B, its accompanying caption, corresponding methods, and main text accordingly.

15. In the sentence “ To recall, the magnitude of the signal contrast is a function of both the effective refractive index and size of the particles, which makes absolute sizing non-trivial and susceptible with errors” the preposition with is inappropriate. Please change accordingly.

Action taken: We have modified the sentence to:

To recall, the magnitude of the signal contrast is a function of both the effective refractive index and size of the particles, which makes absolute sizing non-trivial and susceptible to errors even in the presence of a calibration.

Reviewer #2 (Remarks to the Author):

The manuscript NCOMMS-23-45101-T titled: “Molecular fingerprinting of biological nanoparticles with a label-free optofluidic platform” by Alexia Stollmann and co-authors presents the proof-of-concept of a technology that integrates label-free imaging, microfluidics and surface chemistry at the aim to detect biological nanoparticles. The surface functionalization protocol is based on supported lipid bilayers (SLBs) and the label-free imaging is an in-line digital holographic microscope for the detection of nanoparticle at the limit of diffraction. The work is mainly organized in four parts: (i) Integration of surface functionalization and imaging procedure by defining image processing steps and thresholding to distinguish between background and signal, (ii) Assessments of microfluidic chip reproducibility and robustness (iii) validation of the immunoassay by flowing a sample of streptavidin-labelled 20 nm gold nanoparticles (AuNPs-SAv) and (iv) test of the technology by proving the biosensing performances on six surface biomarkers from 4 different ovarian cancer cell line.

The paper is technically correct and well-written. The data analysis support claims and conclusions and the authors clearly evaluate the limitations of the methodology. The paper is suitable for publication after minor review points will be addressed, specifically:

We thank the reviewer for supporting the publication of our manuscript. The remaining concerns are addressed below.

- In the introduction, the in-line holographic technique is not mentioned. In order to give a complete overview of the proposed method authors should declare it in this introductory section.

All the label-free methods described in the introduction belong to the category of digital inline holographic microscopes. When a laser is used as a light source and the detection is performed in reflection, this subset of inline holographic techniques is commonly referred to as interferometric scattering, also known by its acronym iSCAT. Nonetheless, given the generality of the arguments towards all optical methods that rely on elastic scattering we have kept a rather broad term as “all-optical label-free methods based on elastic scattering”.

Action taken: We have clarified the distinction between inline holographic techniques and elastic-scattering as follows:

From the available all-optical label-free methods, those based on elastic scattering and particularly those belonging to the family of digital inline holography have become one of the most promising, as they now routinely achieve detection sensitivities down to the single protein⁹⁻¹¹, nucleic acid¹², and micelle level¹³ that rival single-molecule fluorescence.

- Concerning the data presented in Fig. 2B, it's not clear how these results are obtained. Authors should add more information on this point.

The data shown in Fig. 2B corresponds to dynamic light scattering (DLS) measurements of different liposome preparations obtained using a commercial DLS instrument.

Action taken: We have added experimental details regarding the DLS measurements in the methods section as follows:

Liposome preparations were diluted and subsequently sized in triplicate using a commercial DLS (Malvern Zetasizer Nanoseries Nano-ZS, T = 25C, 173 backscatter detection)

- In the inset in Figure 2D concerning the background contrast there are different areas, what they refer to? please explain better in the text or in the caption.

Each gray area in Fig. 2D represents the contrast distribution of all particle signatures identified over an estimated scanned substrate area of 0.2 mm^2 . These individual distributions, all having the same gray intensity, are then overlaid with a transparency level effect. Due to the number of experimental replicas being overlaid ($N = 6$), darker areas in the Fig. 2D simply reflect that there is a higher degree of overlap in that region. We have now clarified this and specified the number of replicas for this in the caption.

Action taken. We modified the caption of figure 2D to clarify the source of confusion.

(D) Particle contrast histograms from all localisations found in substrates exposed only to PBS solution ($N = 6$). Each curve with the same transparency level and gray intensity corresponds to an approximate scanned area of 0.2 mm^2 . The overlap between different curves is indicated by the different degrees of transparency.

- Authors should add some information about the computational time of the image processing steps, i.e the time for detecting the spots in a single patch of $100 \times 100 \text{ }\mu\text{m}^2$ and the total computational time due to the stitching of all the patches. In the supporting videos authors declare “real-time” imaging, what they refer to?

All measurements and binding events can be detected in real-time, namely the moment they are recorded. This is possible because the computational time for particle localization is in the range of 10 ms - 100 ms, while the effective time resolution of acquisition is 1 Hz. Nevertheless, all data in the manuscript were batch-processed off-line to avoid introducing biases during the experimental acquisition. In the manuscript we purposely avoid discussing computational time as this is heavily dependent on the computer architecture and on the efficiency of the code, for instance whether this is implemented in CPU or GPU. Given that the assays span minutes to several hours, the computational time does not restrict the analysis or play a significant role. In the supporting videos, real-time refers to the events are observed as they occur.

Regarding the frame stitching, this is simply a visualization tool to determine whether there is any spatial dependence on the density of localized particles within a sensing channel, e.g. Fig. 5D. However, this image processing step is not necessary for either the overall data analysis or the calculation of the particle density, since the sample scans are programmatically controlled with a resolution of 50 nm over a user-defined path.

Action taken. We have added information regarding the total computational in the corresponding methods section.

The resulting lateral position, signal contrast and the integrated signal contrast were stored for further processing. The overall computational time for particle localisation per FOV ranged from 10-100 ms.

- Some sections may be improved in terms of readability. For example in the introduction the concept of coexistence of sensitivity, high-throughput, and molecular fingerprinting is repeated few times; a further example is at page 4 where the description of the imaging setup contains two times the FOV. I suggest to review the manuscript addressing these repetitions.

We thank the reviewer for the constructive feedback to improve the readability of the manuscript.

Action taken. We have polished the entire manuscript by addressing instances of repeatability mentioned as well as making the text more concise. For sake of readability of this response, all changes are highlighted in the main manuscript.

Reviewer #3 (Remarks to the Author):

In the manuscript by Stollmann et al., the authors describe a label-free optofluidic-based approach for fingerprinting biological nanoparticles (i.e., extracellular vesicles) at the single-vesicle level. Specifically, the authors designed a microfluidic platform to capture nanoparticles and nanometric liposomes by implementing the same concepts of an immunoaffinity assay. To bind the nanoparticles to the microchannels, a supported lipid bilayer (SLB) was first formed through rupture and fusion of liposomes at the surface of the microchannel and, then, antibodies were attached to the SLB using NeutrAvidin and biotin. The authors demonstrated that the binding of streptavidin functionalized nanoparticles can be detected at the single particle level and is affected by the particles concentration and flow rate of the dispersion within the microchannel. Finally, the authors demonstrated that the binding profile (i.e., fingerprint) of different ovarian cell line-derived EVs can be measured by using microchannels with different antibodies attached to their surface.

General comments:

In general, the paper by Stollmann et al. nicely describes the development and usage of a label-free optofluidic-based platform for detecting different types of EVs at the single vesicle level. Nevertheless, since this work is a proof of concept study, it mainly focuses on the development and testing of the technique rather than addressing various important challenges in EV detection (e.g., differentiating specific EVs from a sample that contains a mixture of EVs, detection of EVs from real biological samples, etc.). In terms of the experiments described in the paper, it is expected that surface characterization, such as SLB formation and antibodies attachment, will be assessed using other techniques like atomic force microscopy, fluorescence microscopy, etc. Similarly, quantification of binding/dissociation constant and evaluation of various parameters that may influence nanoparticles binding were not reported. This information is critical for determining the optimal conditions (flow rate, nanoparticle concentrations, solution pH, salt concentration, etc.) for nanoparticles binding. Finally, while profiling of EVs using different surface biomarkers was demonstrated as a proof-of-concept, the obtained fingerprints were not significantly different to enable clear differentiation between EV subpopulations.

For these reasons, at this stage the paper by Stollmann et al. does not meet the criteria for publication in Nature Comm.

We regret the reviewer's opinion and appreciate that these concerns were raised to further improve our work. Below, we address the reviewer's concerns point-by-point through a combination of new experiments and further statistical analysis. Nonetheless, we firmly believe the work is suitable for Nat Comms not only due to its relevance, its appeal to a broad readership, but also of the novelty it brings to one of the key pillars of nanotechnology: multiplexed characterisation of heterogeneous nanoparticles samples in a label-free manner.

Throughout the manuscript, we reiterate that this work does not claim nor provide a diagnostic tool based on EV, but rather focusses on showcasing the ability of a label-free platform to discriminate between different nanoparticles using an immunoaffinity pull-down assay that is both compatible with microfluidics and state-of-the-art label-free imaging. We chose to apply

out platform to EVs both because of their high relevance across multiple disciplines, but also because they serve as an ideal model system of highly heterogenous nanoparticles samples that to date remains challenging to characterise.

In terms of the experiments described in the paper, it is expected that surface characterization, such as SLB formation and antibodies attachment, will be assessed using other techniques like atomic force microscopy, fluorescence microscopy, etc.

We respectfully disagree with reviewer's assessment that additional validation steps are needed. Label-free imaging in the form of interferometric scattering is one of the standard methods for SLB characterisation¹, lipid domain visualisation⁹, and detection of proteins on SLBs^{10,11}. Although fluorescence enables fluorescence recovery after photobleaching (FRAP) assays, these assays fail to provide additional information that is relevant, e.g. membrane fluidity, towards the sensitivity and performance of the sensor, i.e. the number of total defects per unit area. We support this claim by performing fluorescence detection of the formed SLB and show that in this imaging modality no defects are detected. With the rather limited area information provided by AFM and the fact that this is not compatible with in-situ characterisation we argue this technique is ill-suited to provide relevant information about the overall quality of the sensor.

Action taken: We have performed complimentary experiments to validate each step of the immunoassay and are now part of the Supplementary Information.

- 1) Fluorescence imaging of the formed SLB doping the lipid composition with fluorescent lipids showcasing that fluorescence imaging is unable to robustly quantify surface defects.
- 2) Label-free imaging of positive and negative controls indicating formation of the SLB with biotinylated lipids. For this we chose streptavidin functionalised 20 nm AuNPs to test the presence of biotinylated lipids as positive control as already shown in Fig. 4B versus EVs as negative control. We further verified the presence of the SLB by showing the number of bound EVs with and without an SLB present.
- 3) Label-free imaging of positive and negative control validating the introduction of Neutravidin. For this we chose the biotinylated liposomes to test the presence of as positive control versus 20 nm AuNPs as a negative control.
- 4) Label-free imaging of positive and negative controls validating the introduction of ABs. For this we chose the streptavidin functionalised 20 nm AuNPs as positive control versus 40 nm carboxyl AuNPs and biotinylated liposomes as negative controls.

Changes to the manuscript:

Fig. S6. Additional surface functionalisation validation. (A) Representative fluorescence image of a formed SLB 0.1% molar doped with a fluorescently-labelled lipid. The homogeneous fluorescence signal shows that no surface defects can be resolved under this imaging modality. Scale bar: 10 μm. (B) Number of bound particles in paired positive (check mark) and negative control (cross) assays to validate different functionalisation steps. (C) Tabulated schematic outlining all the positive and negative control assays reported (Fig. 4C, Fig. S4B, Fig. 6) in the manuscript that validate successful surface functionalisation. Check mark: positive control, cross: negative control, circle: target assay.

Similarly, quantification of binding/dissociation constant and evaluation of various parameters that may influence nanoparticles binding were not reported. This information is critical for determining the optimal conditions (flow rate, nanoparticle concentrations, solution pH, salt concentration, etc.) for nanoparticles binding.

We respectfully disagree with the expectations of the reviewer, which we consider to go well beyond the scope of the current work. In the manuscript, we provide the most significant parameters that affect the binding kinetics, such as flow rate, particle concentration (shown in Fig 4D, 5B) and number of doping sites on the SLB surface (Fig S4). In addition, we provide optimisation guidelines so readers may adjust the platform according to their system, such as changes in the microfluidic design, flow rate and biotin-doping. Moreover, the aim of this work is to generate unique fingerprints based on a panel of surface biomarkers. As such, as long as nanoparticles with the analyte of interest can be immobilised and detected, both the absolute quantification of the binding/dissociation constants and optimization of nanoparticle binding is not required. This is because the fingerprinting hinges on quantifying the relative number of captured nanoparticles given the same experimental conditions, which we accomplish by multiplexed microfluidic integration and situ characterisation of the quality of the sensor.

We consider that going into further parameter optimisation such as solution pH, type of buffer, and salt concentration constitutes a separate work in and of itself. We believe such in-depth characterisation would additionally detract the readers from one of the main points of the

manuscript: showcasing an integrated platform that brings together state-of-the-art label-free imaging with multiplexed microfluidics that can be used to molecularly characterise heterogeneous nanoparticle samples.

Regarding the quantification of the binding/dissociation constants, we specifically refrain from quantifying them because our assay, given the experimental conditions (i.e. a large excess of capture ABs with respect to the analytes) is in the titration regime, which has been extensively reported in the literature to lead to misinterpretations of the K_d values¹². Therefore, reporting values of binding/dissociation constants would only add to the long list of reports that have done so inadequately.

Finally, while profiling of EVs using different surface biomarkers was demonstrated as a proof-of-concept, the obtained fingerprints were not significantly different to enable clear differentiation between EV subpopulations.

By visual inspection, the molecular profiles in Fig. 6D are distinct enough to tell them apart. To further strengthen our argument, we have performed dimensionality reduction via principal component analysis and plotted the respective 2D projection in Fig. 6D. This 2D projection not only shows that the fingerprints are unique amongst EVs from different cell lines, but that a classification between benign and cancerous cell lines is possible as indicated by the different shaded regions separated by the vertical line.

Additional

comments:

1. Page 7: According to the authors, the formation of defect free SLB is critical for minimizing

false positive readouts. Nevertheless, the formation of an intact SLB without defects/boundaries over a large surface area is improbable. Have you characterized the quality of the formed SLB using complementary methods like AFM, fluorescence microscopy, etc.? The reviewer is correct to point out that formation of defect-free SLBs over such large areas is improbable. As a result, throughout the manuscript we refrain from claiming defect-free bilayers. Instead, we consider that these defects are unavoidable and thus should be characterised for each measurement. As such, we devote considerable efforts to systematically assess the conditions that reproducibly minimise the number of defects introduced during the SLB formation and subsequent functionalisation steps (Fig. 2E, 3B, 4A, 4B). As mentioned in the discussion, if these defects are immobile, which they mostly are, and image scans are performed prior to introduction of the sample of interest, then these immobile defects can be referenced, and their false positive contribution minimised if not removed entirely with further image processing.

Although the reviewer suggests other methods to characterise the quality of the bilayer, label-free approaches, such as the one described in this manuscript, have become standard tool that provides the same information as fluorescence. Despite fluorescence enables fluorescence recovery after photobleaching (FRAP) assays, these assays do not provide additional information that is relevant towards the sensitivity and performance of the sensor, i.e. the number of total defects per unit area. On the other hand, AFM provides nanoscopic information inaccessible to optical-based techniques, yet the overall imaging area is rather limited to areas in the range of $10 \times 10 \text{ um}^2$, and not compatible with directly measuring in-situ, i.e. within the microfluidic environment. Ultimately, despite the SLB quality can be assessed by other methods, our approach provides an in-situ quantitative characterisation of the very same sensor area that is being used for the fingerprinting assay; thus, enabling us to account for slight differences in SLB quality between experiments. These differences in the final functionalised SLB quality can stem from different either intrinsic coverslip substrate defects, variations in chip fabrication, leaching of PDMS oligomers, aggregation of antibodies, introduction of contaminants, etc..., which simply would not be measured via other approaches.

Action taken: We have included the following statements that the label-free imaging approach allows us to in-situ characterise the quality of each sensing channel.

Furthermore, we opted for label-free imaging over other complementary approaches such as fluorescence and AFM, because it provides an in-situ quantitative characterisation of the very same sensor area at each stage of the immunoassay, which simply would not be measured via other approaches; thus, enabling us to account for slight differences in SLB quality between experiments.

2. Page 11 and figure 5A: Why the assay takes so long (hours)? Have you tested different flow rates except from the ones you reported? What about the density of antibodies at the SLB surface? the paper reports a 99:1 molar ratio, why not higher than this? Are the antibodies/proteins washed away at higher flow rates? Regarding the assay time, we explicitly make the reader aware that this is slower compared to other approaches, and have discussed in detail the reason for this in pages 12 and 13 of the main text as a combination of mass transport limited reaction, EV avidity and sensor attainability.

Regarding the flow rates, we tested the range of flow rates that our pressure controller could deliver at the time.

Regarding the density of antibodies, we provide back-of-the-envelope calculations of the expected surface density and effect concentration in the channel in page 12: “We specifically computed the capture antibodies to be in the μM range given an estimated density of 0.08 pmol/cm^2 and microfluidic channel height of $10 \mu\text{m}$.” We perform these estimations based on the molar percentage of biotin doping which results in a surface area of 1.4 biotins per $10 \times 10 \text{ nm}^2$ (2.3 pmol/cm^2), which translate into an almost full monolayer of NeutrAvidin. Note that a full NeutrAvidin monolayer occurs at biotin densities of 2.8 biotins per $10 \times 10 \text{ nm}^2$ (3.5% molar biotin, 8 pmol/cm^2)¹³. Furthermore, considering that ABs have a larger surface area than NeutrAvidin and assume the maximum coverage of one AB per available biotin on the SLB, we obtain a surface density of 1 AB per $10 \times 10 \text{ nm}^2$ or 10,000 ABs/ μm^2 . These back-of-the-envelope calculations are based on prior work that specifically studied streptavidin coverage on biotinylated surfaces; from which a predictive model was derived to quantify the surface density of streptavidin from the molar biotin doping. Given the larger surface area and that neutravidin is expected to form a full monolayer at 3.5 molar doping, we argue there is no reason to go to any biotin doping ratios higher than 99:1 molar.

Regarding whether higher flow rates wash away the ABs or proteins, we argue there is no evidence to suggest this occurs given the much higher number of EVs immobilised to the functionalised surface at higher flow rates (Fig. 5C). Moreover, we would expect the flow to tear the bilayer well before the dissociation between the biotin-Neutravidin ($K_d \approx 10^{-14} \text{ M}$) moieties. Since we do not see the bilayer tear at the flow rates applied, we rule out any contribution from the ABs washing away.

3. Page 11: are the testes EV concentrations comparable to biological samples?

In this work we targeted an EV concentration range between $1\text{e}9$ - $1\text{e}10$ EVs/mL, which is a range routinely accessible by different isolation techniques and falls within the range of some biologically relevant concentration of EVs which can range from $1\text{e}6$ - $1\text{e}10$ EVs/ml, dependent on the biological fluid or physiological state of the individual. We state in the manuscript that the assay was optimized for a concentration range between $1\text{e}9$ - $1\text{e}10$ EVs/mL, and purportedly do not claim absolute concentration quantitation, but rather the relative expression levels amongst the panel of biomarkers, which is sufficient for generating a fingerprint.

Furthermore, in the Limitation section of the manuscript we state that work is still required to improve the passivation against non-specific binding for the method to be applied to direct biological fluids. That said, the parameters of the chip can be adjusted to the address lower or higher concentration ranges, or even achieve faster kinetics. We would like to stress that the point of the paper is to present a method that interfaces state-of-the-art label-free characterisation with microfluidics to generate a unique molecular fingerprint. We showcase the multiplexed molecular fingerprinting using extracellular vesicles as a model system for heterogeneous biological nanoparticle samples, and present an example of its potential application.

4. In page 11 you mention “...to retrieve a reliable dissociation rate constant...”. I could find any estimation of the dissociation rate constant in the paper.

We apologize for the confusion. What we originally meant was that we could not obtain the dissociation rate constant as EVs barely detached from the chip. We have revised the sentence for clarification.

Action taken: We have modified the sentence to:

In contrast, we were unable to retrieve a reliable dissociation rate constant, k_{off} , as EV unbinding events were barely detected during the buffer exchange (Fig. 5A inset).

5. Figure 5A (time lapse images): time should be indicated in the image. Each image from the time-lapse panel corresponds to a time-point from the kinetic curve except for $t=0$.

Action taken: We have indicated the time for each of the images in the time-lapse panel to avoid confusion.

6. Figure 6: A major challenge in this area is developing methodologies that can differentiate the different types of EVs in a given biological sample. The work presented by the authors does not show this kind of ability. Can the method presented by the authors be utilized for detecting various EV subpopulations in a given biological sample?

We note that our platform can differentiate EVs based on the expression levels of a panel of surface markers, which we then use as a classifier to determine whether the EVs originate from either a benign or cancerous cell line. A similar approach using aptamers and NanoFCM already showed that such classification based on surface biomarkers is extremely promising for diagnosis¹⁴. Therefore, we believe that further work to combine information based on biochemical and physical properties (e.g. particle size and scattering properties), as well as development of better surface chemistry to work with patient-derived EVs, both of which lie beyond the scope of this work, will significantly improve the classification and extend the platforms use to diagnostic applications.

7. Figure 6D: The counts for the ovarian cancer biomarkers is very low relative to the classical biomarkers which appear for all types of EVs. To differentiate between the different populations of EVs a significantly different binding profile is expected to appear for the cancer biomarkers.

Recent studies¹⁵⁻¹⁸ have shown that the expressions levels of classical tetraspanin EV markers (CD9, CD81, CD63) routinely exceed those of other markers, including cancer markers. Furthermore tetraspanins alone have the potential to differentiate between EVs from different sources¹⁹. Our data show that the expression profile of cancer-markers is sufficiently different among EVs from cancerous and benign cells (see new analysis in Fig. 6D). These results support the potential our system for EV-based cancer detection. However, we do acknowledge the further need for validation with a larger number of samples, both in terms of experimental replicas and from different cell line sources, which we discussed in the Limitation section of the manuscript.

Action taken: We have added the following discussion into the Limitation section.

In this work we used a small selection of EV samples and experimental replicas as a proof-of-concept experiment to generate unique fingerprints. Nevertheless, to extend this work from mere EV population differentiation into a robust classification library⁶¹, further validations are required. Specifically, from a statistical perspective there is need to increase the number of experimental replicates and the number of different EV samples. Similarly, the number of required biomarkers in the fingerprint to generate a robust classification should be further investigated.

REFERENCES

1. Andrecka, J., Spillane, K. M., Ortega-Arroyo, J. & Kukura, P. Direct Observation and Control of Supported Lipid Bilayer Formation with Interferometric Scattering Microscopy. *ACS Nano* **7**, 10662–10670 (2013).
2. Cho, N. J., Hwang, L. Y., Solandt, J. J. R. & Frank, C. W. Comparison of extruded and sonicated vesicles for planar bilayer self-assembly. *Materials* **6**, 3294–3308 (2013).
3. Zhu, L., Gregurec, D. & Reviakine, I. Nanoscale departures: Excess lipid leaving the surface during supported lipid bilayer formation. *Langmuir* **29**, 15283–15292 (2013).
4. Spillane, K. M. *et al.* High-speed single-particle tracking of gm1 in model membranes reveals anomalous diffusion due to interleaflet coupling and molecular pinning. *Nano Lett* **14**, 5390–5397 (2014).
5. Ortiz-Orruño, U., Jo, A., Lee, H., Van Hulst, N. F. & Liebel, M. Precise Nanosizing with High Dynamic Range Holography. *Nano Lett* **21**, 317–322 (2021).
6. Huang, Y. F. *et al.* Coherent Brightfield Microscopy Provides the Spatiotemporal Resolution to Study Early Stage Viral Infection in Live Cells. *ACS Nano* **11**, 2575–2585 (2017).
7. Holanová, K., Vala, M. & Piliarik, M. Optical imaging and localization of prospective scattering labels smaller than a single protein. *Opt Laser Technol* **109**, 323–327 (2019).
8. Feliu, N., Sun, X., Alvarez Puebla, R. A. & Parak, W. J. Quantitative Particle-Cell Interaction: Some Basic Physicochemical Pitfalls. *Langmuir* **33**, 6639–6646 (2017).
9. de Wit, G., Danial, J. S. H., Kukura, P. & Wallace, M. I. Dynamic label-free imaging of lipid nanodomains. *Proceedings of the National Academy of Sciences* **112**, 12299–12303 (2015).
10. Heermann, T., Steiert, F., Ramm, B., Hundt, N. & Schwille, P. Mass-sensitive particle tracking to elucidate the membrane-associated MinDE reaction cycle. *Nature Methods* **18**, 1239–1246 (2021).
11. Foley, E. D. B., Kushwah, M. S., Young, G. & Kukura, P. Mass photometry enables label-free tracking and mass measurement of single proteins on lipid bilayers. *Nature Methods* **18**, 1247–1252 (2021).
12. Jarmoskaite, I., Alsadhan, I., Vaidyanathan, P. P. & Herschlag, D. How to measure and evaluate binding affinities. *Elife* **9**, 1–34 (2020).

13. Hamming, P. H. E. & Huskens, J. Streptavidin Coverage on Biotinylated Surfaces. *ACS Appl Mater Interfaces* **13**, 58114–58123 (2021).
14. Li, J. *et al.* An Aptamer-Based Nanoflow Cytometry Method for the Molecular Detection and Classification of Ovarian Cancers through Profiling of Tumor Markers on Small Extracellular Vesicles. *Angewandte Chemie International Edition* **63**, (2024).
15. Lee, K. *et al.* Multiplexed Profiling of Single Extracellular Vesicles. *ACS Nano* **12**, 494–503 (2018).
16. Nikoloff, J. M., Saucedo-espinoza, M. A. & Dittrich, P. S. Microfluidic Platform for Profiling of Extracellular Vesicles from Single Breast Cancer Cells. (2022)
doi:10.1021/acs.analchem.2c04106.
17. Spitzberg, J. D. *et al.* Multiplexed analysis of EV reveals specific biomarker composition with diagnostic impact. *Nat Commun* **14**, 1239 (2023).
18. Kwak, T. J. *et al.* Electrokinetically enhanced label-free plasmonic sensing for rapid detection of tumor-derived extracellular vesicles. *Biosens Bioelectron* **237**, 115422 (2023).
19. Mizenko, R. R. *et al.* Tetraspanins are unevenly distributed across single extracellular vesicles and bias sensitivity to multiplexed cancer biomarkers. *J Nanobiotechnology* **19**, 1–17 (2021).

Reviewers' Comments:

Reviewer #1:

Remarks to the Author:

The authors have substantially and adequately addressed comments/criticisms made of the original script. Nevertheless, to ensure the readiness of the manuscript for publication in Nat. Commun., there are a couple of minor points that merit attention:

1. Further iterations appear warranted to fully establish the robustness of the classification library as asserted by the authors. It is recommended, to conduct a minimum of N=3 iterations in Figure 6 to strengthen the reliability and comprehensiveness of the findings.
2. Given that the underlying reaction mechanism of the proposed system hinges upon antigen-antibody binding interactions, it is imperative to broaden the investigation scope. Specifically, in addition to targeting antigens present on the surface of exosomes, it is necessary to explore the signal when a single protein is spiked onto the assay.

Reviewer #2:

Remarks to the Author:

Concerning the manuscript NCOMMS-23-45101A, authors properly addressed comments and suggestions. The paper is suitable for publication in Nature Communications Journal

Reviewer #3:

Remarks to the Author:

Following the revisions done by authors, the paper by Stollmann et al. is suitable for publication in Nature Comm.

REPLY TO REVIEWERS: NCOMMS-23-45101-A

REVIEWER

COMMENTS

Reviewer #1 (Remarks to the Author):

The authors have substantially and adequately addressed comments/criticisms made of the original script. Nevertheless, to ensure the readiness of the manuscript for publication in Nat. Commun., there are a couple of minor points that merit attention:

We thank the reviewer for taking the time to evaluate our manuscript and providing constructive feedback to improve our work. Below, we address the concerns point-by-point.

1. Further iterations appear warranted to fully establish the robustness of the classification library as asserted by the authors. It is recommended, to conduct a minimum of N=3 iterations in Figure 6 to strengthen the reliability and comprehensiveness of the findings.

For all fingerprinting assays we performed a minimum of 3 chip replicates. However, due to either the presence of defect along one of the biomarker channels or to higher levels of non-specific binding, we excluded the chip replicates from the analysis. Further to keep the number of replicates consistent throughout the fingerprinting we chose the minimum number available for all cell lines (N = 2). By correcting for different levels of non-specific binding and by normalising to the average expression levels of the three pan-EV markers (CD9,CD81,CD63) we were able to showcase all experimental replicates, including those that were initially excluded.

Action taken: We have incorporated replicates that had been initially excluded due to higher levels of non-specific binding, single channel defect and to compare the same number of replicates. The text referencing Figure 6 and its caption have been modified accordingly.

To determine whether the fingerprints are unique enough to differentiate amongst different EV populations, we repeated this measurement with a minimum of 3 chip replicas for all EV samples. To correct for different levels of non-specific binding, the average count density of the negative control was subtracted from each surface marker on a chip-by-chip basis. Then to correct for differences in EV concentration and variations in flow rate between chips and EV samples, each fingerprint was normalised to the average expression level of the three tetraspanin markers (Fig. 6D). We opted for such normalisation to determine which biomarkers were positively expressed above the level given by the non-specific binding and for the fingerprints to be independent of EV concentration and flow rate. Overall, the pan-EV tetraspanin markers showed higher expression levels compared to other markers, in agreement with other recent works^{3-5,48}, and displayed visible differences between the EVs populations, which have been shown to be sufficient to differentiate between EVs from different cell lines⁷. For the benign cell line TiOSE4 and the cancer ES2, only the pan-EV tetraspanins were positively detected⁵⁷, with the cancer biomarkers showing the same expression levels as the negative control. In contrast, EVs from the cancerous cell lines OV90 and CaOV3 showed positive expression levels for all cancer biomarkers to varying degrees; yet, both EV populations followed a general surface protein expression trend of CD326 >> HE4 > CA125. Combining the pan-EV tetraspanins markers with the cancer specific ones and performing a dimensionality reduction via principal component analysis (PCA) confirmed that the EV fingerprints from each ovarian cell line are unique and

could be differentiated as indicated in the 2D projection (Fig. 6D). An exception occurred for one of the fingerprints of OV90, light purple point in PCA 2D projection, where a defect in the CD63 channel affected the overall flow and thus binding kinetics. This defect resulted in a much lower expression level of CD63 compared to the other 3 chip replicates. Furthermore, such dimensionality reduction also differentiated between EVs originating from benign and cancerous cell lines as indicated by the vertical line separating the two different shaded regions; thus, demonstrating the potential of the platform to characterise different subpopulations expressing specific target analytes within a highly heterogenous nanoparticle sample.

Fig. 6. Multiplexed EV fingerprinting. (A) Schematic representation of the multiplexed in-chip immunoaffinity assay used to profile extracellular vesicles expressing different surface markers. Different colours encode the antibody used to target a specific surface marker. Each sensing channel was independently functionalised with a different capture antibody. (B) In-chip binding kinetics CaOV3 EVs expressing their respective surface protein markers. (C) Molecular fingerprint of CaOV3 EVs expressed in terms of the number of EVs captured within each channel after 5h of continuous flow. The captured vesicles are given in both total and normalised incidences. Data are expressed as mean \pm SD over independent chip replicates ($N = 4$). (D) Normalised molecular fingerprint of four different ovarian cell line-derived EVs alongside its corresponding 2D projection after dimensionality reduction with principal component analysis (PCA). Data are expressed as mean \pm SD over independent chip replicates ($N = 4, 4, 3, 3$). Vertical line in the 2D projection separates EVs cell lines into cancerous (orange area) and benign (blue area) types. (E) Representative contrast distribution for all EVs captured within a single CaOV3 EV fingerprinting assay. Shaded area indicates the background contrast region that is not considered in the EV count density metric. Inset: correlative scatterplot comparing the median contrast magnitude against the count density of each marker for independent chip replicates of CaOV3 EVs ($N = 4$).

2. Given that the underlying reaction mechanism of the proposed system hinges upon antigen-antibody binding interactions, it is imperative to broaden the investigation scope. Specifically,

in addition to targeting antigens present on the surface of exosomes, it is necessary to explore the signal when a single protein is spiked onto the assay.

We believe that this point goes well beyond the scope of the current manuscript. The reviewer is correct to point out that immunoaffinity pull-down relies on antibody-antigen interactions; however, the analytes we target are membrane-associated proteins specifically present on the outer surface of EVs. These analytes would not otherwise be found as free protein, so they would not affect the signal of the sensor whatsoever.

If the reviewer refers to the influence of non-specific interactions between free protein and EVs, then this is precisely the reason why we always perform an isotype control using IgG1 antibodies for each chip replicate -- with the specific purpose to correct for the levels of non-specific binding. In other words, we use this negative control to quantify and subsequently correct for non-specific antigen-antibody interactions. See fig. 6 and discussion above.

Finally, even if we target specific proteins that are present freely in solution (i.e. not associated to EVs) the signals from single proteins are at least one order of magnitude smaller than those for EVs and the roughness of the surface functionalisation. Thus, to detect these signals we would have to perform differential imaging, an modality that was not used in this work for EV fingerprinting.

Action taken: To demonstrate that these signals are much smaller and that our immunoassay can also be applied to single proteins we performed immunoaffinity pulldown assays against spiked fibronectin (220 kDa) in a landing assay (microwell) and under continuous flow (in chip). The signals recovered from single fibronectin particles are at least 10x smaller than the median contrast of the EVs.

Immuno-assay applied to free protein in solution. (A) Schematic of the two immunoaffinity capture assays and corresponding single protein binding images showing single particle binding (pink ring) and unbinding (white ring) events. Top: on-chip continuous flow assay. Bottom: microwell landing assay. **(B)** Contrast histogram detailing each binding and unbinding event. The continuous flow contrast histogram has been scaled down for comparison. Vertical lines and associated numerical values indicate the contrast of the population average. Right: Localisation map detailing each fibronectin binding and unbinding event in the on-chip continuous flow (top) and landing (bottom) assays, respectively. Scale bars: 10 μm .